



# Climate change will cause non-analog vegetation states in Africa and commit vegetation to long-term change

**Mirjam Pfeiffer**[1], **Dushyant Kumar**[1], **Carola Martens**[1,2], and **Simon Scheiter**[1]

[1]Senckenberg Biodiversity and Climate Research Centre (BiK-F), Senckenberganlage 25,
60325 Frankfurt am Main, Germany
[2]Institute of Physical Geography, Goethe University Frankfurt am Main, Altenhoeferallee 1,
60438 Frankfurt am Main, Germany

**Correspondence:** Mirjam Pfeiffer (mirjam.pfeiffer@senckenberg.de)

**Abstract.** Vegetation responses to changes in environmental drivers can be subject to temporal lags. This implies that vegetation is committed to future changes once environmental drivers stabilize; e.g., changes in physiological processes, structural changes, and changes in vegetation composition and disturbance regimes may happen with substantial delay after a change in forcing has occurred. Understanding the trajectories of such committed changes is important as they affect future carbon storage, vegetation structure, and community composition and therefore need consideration in conservation management. In this study, we investigate whether transient vegetation states can be represented by a time-shifted trajectory of equilibrium vegetation states or whether they are vegetation states without analog in conceivable equilibrium states. We use a dynamic vegetation model, the aDGVM (adaptive Dynamic Global Vegetation Model), to assess deviations between simulated transient and equilibrium vegetation states in Africa between 1970 and 2099 for the RCP4.5 and 8.5 scenarios using regionally downscaled climatology based on the MPI-ESM output for CMIP5. We determined lag times and dissimilarity between simulated equilibrium and transient vegetation states based on the combined difference of nine selected state variables using Euclidean distance as a measure for that difference. We found that transient vegetation states over time increasingly deviated from equilibrium states in both RCP scenarios but that the deviation was more pronounced in RCP8.5 during the second half of the 21st century. Trajectories of transient vegetation change did not follow a "virtual trajectory" of equilibrium states but represented non-analog composite states resulting from multiple lags with respect to vegetation processes and composition. Lag times between transient and most similar equilibrium vegetation states increased over time and were most pronounced in savanna and woodland areas, where disequilibrium in savanna tree cover frequently acted as the main driver of dissimilarities. Fire additionally enhanced lag times and dissimilarity between transient and equilibrium vegetation states due to its restraining effect on vegetation succession. Long lag times can be indicative of high rates of change in environmental drivers, of metastability and non-analog vegetation states, and of augmented risk for future tipping points. For long-term planning, conservation managers should therefore strongly focus on areas where such long lag times and high residual dissimilarity between most similar transient and equilibrium vegetation states have been simulated. Particularly in such areas, conservation efforts need to consider that observed vegetation may continue to change substantially after stabilization of external environmental drivers.

## 1 Introduction

Vegetation dynamics is influenced by a variety of environmental drivers, including climatic conditions, atmospheric $CO_2$ concentration, soil parameters, nutrient availability, and disturbance regime (Eamus et al., 2016). These environmental drivers affect vegetation processes on a variety of levels, from physiological processes at the leaf level to community assembly processes at ecosystem level (Felton and

Smith, 2017), and ultimately determine large-scale vegetation patterns on biome level (Lavergne et al., 2010; Woodward et al., 2004). The impact of environmental drivers is reflected in vegetation structure, vegetation-related ecosystem functions, and biogeochemical processes such as carbon sequestration, nutrient turnover, and ecohydraulics (Bonan, 2019). Although environmental drivers are subject to constant variation, vegetation response does not happen instantaneously in accordance with forcing but requires time to allow the system to respond (Essl et al., 2015a). It can therefore be expected that climate change will cause widespread shifts in the distribution of major vegetation formations until the end of the century (Lucht et al., 2006). How much time vegetation requires to respond depends on (i) the type of process that is affected, (ii) the extent of change in the environmental driver, and (iii) the velocity of change, i.e., how fast the driver changes. For example, physiological processes at the leaf level can adapt to changing environmental drivers such as temperature on very short (sub-)daily timescales (Chen et al., 1999; Vico et al., 2019), whereas adaptation to climate change at community level can require years to decades. Slow gradual changes allow vegetation more reaction time, whereas rapid changes leave vegetation drastically behind (Davis, 1989; Corlett and Westcott, 2013). Continuous fluctuation of environmental drivers entails that vegetation is usually not in equilibrium with forcing at a given time, and disequilibrium vegetation dynamics under future climate change needs to be expected (Svenning and Sandel, 2013).

Temporal lags between forcing and vegetation state imply that vegetation is committed to further changes even if environmental drivers stabilize (Jones et al., 2009; Scheiter et al., 2020). It is particularly important to consider this when estimating or mitigating the effects of future climate change. Committed vegetation changes at the time of stabilization of climatic drivers have important implications for carbon storage (Pugh et al., 2018), vegetation structure, and community composition. In addition, delayed responses to environmental drivers may unexpectedly push vegetation beyond tipping points towards alternative stable states long after the change in forcing has occurred. Particularly in connection with African savanna ecosystems, such multi-stable ecosystem states have been proposed and studied by a variety of authors (e.g., Staal et al., 2016; Li et al., 2019; Pausas and Bond, 2020). Conservation management needs to be aware that the vegetation state at any given time may not be the vegetation state expected under prevailing environmental conditions, and managers need to decide whether to preserve the status quo or allow vegetation development towards its anticipated equilibrium state. Otherwise, climatic disequilibrium may severely threaten the conservation of priority ecosystems (Huntley et al., 2018).

Estimating vegetation trajectories and lags is challenging, and only few studies take into account that plant community changes could substantially lag behind climatic changes (Alexander et al., 2017). This is true when considering changes in single environmental drivers and becomes increasingly complex when considering concurrent changes in multiple drivers. In a previous study, we examined how $CO_2$ concentration change over a range from 100 to 1000 ppm, at two different rates, affects African vegetation and vegetation lags with respect to equilibrium states using the aDGVM (adaptive Dynamic Global Vegetation Model, Scheiter et al., 2020). In that study, we found substantial deviances and lags between equilibrium and transient vegetation states when we increased or decreased $CO_2$. However, in this previous study we only considered $CO_2$ effects while keeping long-term averages of other environmental drivers of vegetation, such as precipitation and temperature, constant. While an estimate of the effect of $CO_2$ in isolation is valuable, a more accurate assessment of lags, debt and surplus in carbon, vegetation cover, and vegetation structure additionally requires consideration of climatic drivers. This is particularly relevant when addressing committed vegetation change for future scenarios of climate change, e.g., the climate change associated with the RCP (Representative Concentration Pathway, Meinshausen et al., 2011) scenarios.

Moreover, when considering multiple drivers of vegetation dynamics, complexity increases. The combination of different drivers may amplify (if they act in the same direction) or weaken (if they act in opposing directions) effects on vegetation when compared to single-driver scenarios. For example, $CO_2$ fertilization effects may be reduced by other factors that inhibit plant growth, such as nutrient limitation or increased water stress. Elevated $CO_2$ is often linked to higher water use efficiency in $C_3$ plants. However, this effect seems to have its limits, and $CO_2$ fertilization cannot always counteract the effects of reduced water availability (Temme et al., 2019). Future changes in precipitation regime, e.g., in precipitation seasonality (prolonged dry season duration), combined with changes in precipitation frequency distribution and annual quantities, are very likely and already observed in different parts of Africa (Batisani and Yarnal, 2010; Dunning et al., 2018). Where water stress increases due to higher drought frequency and severity or changes in precipitation seasonality, its negative effects may be too strong to be offset by $CO_2$ fertilization (see, e.g., Jin et al., 2017; Liu et al., 2020). For a realistic evaluation of vegetation lags associated with future climate change, it is therefore necessary to assess the coupled effects of different drivers in the climate system.

An open question that conservation managers as well as vegetation modelers need to consider is whether observable transient vegetation states correspond to conceivable equilibrium states or whether non-analog vegetation states exist, i.e., vegetation states that have no corresponding equivalent in vegetation states of the past or present. Two possible scenarios are conceivable. In scenario (1), transient vegetation dynamics follows a virtual trajectory defined by equilibrium states. Vegetation lags simply correspond to a time shift of equilibrium states that should exist at a given time according to prevailing environmental conditions; i.e., transient vegeta-

tion states are analog to equilibrium vegetation states of another point in time. In scenario (2), transient vegetation states have no exact analog in any conceivable equilibrium states; i.e., transient vegetation states not only lag behind an equilibrium, but are also "chimeras" that can never be represented by an equilibrium vegetation state. Such mixed vegetation states that entirely lack accordance with any conceivable equilibrium vegetation states are what we define as "non-analog" in the context of this study. Scenario (2) may result from mismatches between equilibrium and transient states at different levels of plant- and vegetation-related processes. As all these processes operate at different timescales, the time lag between various transient state variables and their respective equilibria at any given time will differ, resulting in vegetation disequilibrium with respect to multiple variables. Scenario (2) has important implications, as the complexity of disequilibrium in this scenario constitutes a major challenge for future conservation efforts (Svenning and Sandel, 2013).

Here, we used the aDGVM to assess deviations between transient and equilibrium vegetation states in Africa. The aDGVM has been developed with specific focus on savannas and tropical vegetation, and its performance has been evaluated in a number of studies. In this study, we use the model to compare transient and equilibrium vegetation states in Africa between 1970 and 2099 for RCP4.5 and RCP8.5 on a decadal basis. Using projected climate and $CO_2$ concentrations of the RCPs allows evaluation of the combined effects caused by simultaneous variation of several drivers of vegetation dynamics. We asked the following.

1. How do simulated transient vegetation states deviate from equilibrium vegetation states expected under given historic and future climate conditions, with respect to ecosystem variables related to biomass, vegetation structure, and composition?

2. Do trajectories of transient vegetation change follow a "virtual trajectory" of analog equilibrium states or are transient vegetation states non-analog and different from any equilibrium vegetation state?

3. What are the lag times between transient and most similar equilibrium vegetation states, and which state variables and underlying processes can explain dissimilarities?

4. Which biomes and regions in Africa are most resistant to climate change, and which ones are most prone to experiencing meta-stability and change as a consequence of changing environmental drivers in the future?

## 2 Methods

### 2.1 Model description

The aDGVM (Scheiter and Higgins, 2009) has been developed with emphasis on grass–tree interactions in tropical ecosystems. Trees are simulated as single individuals, and the model incorporates an individual-based representation of plant physiological processes and allows dynamic adjustment of leaf phenology and carbon allocation to environmental conditions. Carbon investment to biomass pools adjusts dynamically in such a way that allocation to those biomass pools that are the most limiting factor for plant growth at a given time is maximized. For example, if water is limiting, more carbon is allocated to roots at the expense of allocation to stems and leaves to increase water uptake capacity, whereas under light limitation, more carbon is allocated to stems and/or leaves to increase light capture. State variables such as biomass, height, and photosynthetic rates keep track of plant performance, while external disturbances such as herbivory (Scheiter and Higgins, 2012), fire (Scheiter and Higgins, 2009), and land use (Scheiter and Savadogo, 2016; Scheiter et al., 2019) impact plants as a function of their traits. The aDGVM simulates four plant types (Scheiter et al., 2012): fire-sensitive but shade-tolerant forest trees, fire-tolerant but shade-intolerant savanna trees, $C_3$ grasses, and $C_4$ grasses, with each type of grass being represented by two types of super-individuals that distinguish grasses beneath or between tree canopies. Physiological differences between $C_3$ and $C_4$ photosynthesis distinguish $C_3$ and $C_4$ grasses and their performance under specific environmental conditions (e.g., Taylor et al., 2018). Fire is modeled as a function of fuel loads, fuel moisture, and wind speed (Higgins et al., 2008) and ignitions are based on a random sequence. It removes aboveground grass biomass and affects trees based on fire intensity and tree height (Higgins et al., 2000, topkill effect). Large trees with crowns above the flaming zone are largely fire-resistant, and grasses and topkilled trees can regrow from root reserves after fire (Bond and Midgley, 2001). Mortality in the aDGVM is a probabilistic function of negative carbon balance. Scheiter and Higgins (2009) and Scheiter et al. (2012) showed that the aDGVM captures the distribution of major vegetation formations in Africa. Scheiter and Higgins (2009) showed that the aDGVM can simulate biomass dynamics in a long-term fire manipulation experiment in Kruger National Park (experimental burn plots, Higgins et al., 2007), and Scheiter and Savadogo (2016) showed that an adjusted model version can reproduce grass biomass and tree basal area under various grazing, harvesting, and fire treatments in Burkina Faso. Scheiter and Higgins (2009) and Scheiter et al. (2015) showed that the aDGVM can simulate broad patterns of fire activity in Africa and Australia, respectively. For a more detailed description of the aDGVM, see Scheiter and Higgins (2009).

## 2.2 Climate forcing data

Simulation of transient vegetation dynamics required time series of climate data. In this study, we used daily climate data that were downscaled with the variable-resolution conformal-cubic atmospheric model (CCAM, McGregor, 2005) for Africa for the period between 1970 and 2099. The downscaling was performed by the South African research group Climate Studies, Modelling and Environmental Health at the Council for Scientific and Industrial Research (CSIR) (Archer et al., 2018; Davis-Reddy et al., 2017; Engelbrecht et al., 2015). The downscaling used GCM projections from the Coupled Model Intercomparison Project Phase 5 (CMIP5, Table S1, IPCC, 2013) and followed the methodology described in Engelbrecht et al. (2015), applying CCAM globally at a quasi-uniform resolution of approx. 50 km in the horizontal. Bias correction of downscaled climate data was performed based on monthly climatologies of temperature and rainfall from CRU TS3.1 data for the period 1961–1990 following Engelbrecht et al. (2015) and Engelbrecht and Engelbrecht (2016). CCAM output is available at daily temporal resolution on a latitude–longitude grid of 0.5° resolution for RCP4.5 and RCP8.5. RCP4.5 is a modest–high-impact scenario with peaking greenhouse gas emissions around mid-century and a $CO_2$ concentration of ca. 540 ppm in 2100. In the high-emission RCP8.5 scenario, emissions keep rising to the end of the century, where $CO_2$ concentrations will reach ca. 900 ppm. Climate variables used in aDGVM simulations were precipitation, daily minimum and maximum temperature, wind speed, and relative humidity. As projected radiation was not available from CCAM, it was derived based on sunshine percentage (Allen et al., 1998) from the New et al. (2002) dataset.

## 2.3 Experimental design

For our simulations, we used CCAM downscaled climate data for RCP4.5 and RCP8.5 based on the boundary conditions provided by the Max Planck Institute Earth System Model (MPI-ESM, Giorgetta et al., 2013). To obtain equilibrium vegetation states on a decadal basis, we conducted separate simulations for all decades between 1970 and 2099; i.e., 13 decadal equilibrium runs per RCP scenario were performed. For each decade, a 250-year random sequence of yearly climate data was generated using the respective RCP scenario's climate data for that decade. In order to avoid a saw-tooth pattern caused by potential small intradecadal trends in climate, the yearly climate forcing for the spin-up of the transient runs and the equilibrium simulations was assembled as a random sequence of the annual climates for the years within a given decade; i.e., the climate of a respective decade was split into 10 annual blocks, which were then randomly put together to create the 250-year climate sequence. The resulting randomized 250 years of climate data were used for equilibrium simulations allowing modeled variables

to reach steady state with the environmental conditions of the decade. Previous simulations have shown that after 200–250 simulation years, the aDGVM reaches equilibrium state for large parts of the study region. The last 30 years of the 13 equilibrium runs were used to determine equilibrium vegetation states for each RCP scenario. The decadal equilibrium states provided the reference base for comparison with decadal results from the transient simulations.

For transient simulations, a 210-year model spin-up was performed using randomly generated sequences of the years in the period 1970 to 1979 to ensure steady-state conditions. After model spin-up, the aDGVM was then forced with the respective RCP climate time series for the period 1970 to 2099 to obtain simulation results of the transient vegetation state. All simulations were conducted both in the presence and absence of fire; i.e., in total eight simulation scenarios were conducted, amounting to a total of 56 simulation runs (4 transient runs, $4 \times 13$ equilibrium runs). Transient model runs were conducted previously by Martens et al. (2020).

## 2.4 Analyses

Comparison of equilibrium and transient vegetation states was conducted using decadal averages of selected state variables at grid cell level that were calculated from annual maximum values (grass and tree biomass) or annual average values. Model variables under consideration were aboveground tree biomass, aboveground grass biomass, savanna tree cover, forest tree cover, total tree cover, average tree height, maximum tree height, number of tree individuals, and the $C_3 : C_4$ grass ratio based on respective totals of grass leaf biomass. Decadal averages for equilibrium scenarios were calculated from the last 30 years of the 250-year simulation sequence. For transient simulations, decadal averages were calculated based on annual simulation output for the respective decades. Although all analyses in this study were conducted on a decadal basis, we focus on three decades (2010s, 2050s, 2090s) in the results section. Full sets of maps for all decades from 1970 to 2099 are provided as video sequences in the Supplement of this study.

### 2.4.1 Comparison between scenarios

Scenarios were compared individually for each key variable to address question 1, i.e., to determine how simulated transient vegetation states deviate from equilibrium vegetation states with respect to specific ecosystem state variables. We calculated continental-scale averages of each key variable based on grid cell values of decadal variable averages and plotted the result as time series.

We calculated the Euclidean distance between transient and equilibrium vegetation states to evaluate the similarity between these scenarios on a per-grid cell and per-decade basis, in order to address question 2. As the nine key variables used for the calculation of Euclidean distance differed in

units and value ranges, we standardized all variables based on variable mean and standard deviation across all decades, grid cells, and scenarios. The standardization across all decades and grid cells of all scenarios to a common mean allows comparison of distance values between scenarios.

Euclidean distance was calculated between same-decade partners (SDPs) in transient and equilibrium simulations to determine the development of similarity over time. To answer question 3, for each transient decade the Euclidean distance to all previous equilibrium decades was calculated, and the equilibrium decade with the closest distance to the respective transient decade was assigned as the closest-decade partner (CDP). We denote the time difference between closest-decade partners as "lag time" in the wider sense, i.e., not taking into account the residual distance between closest-decade partners. This distance should be close to zero for a definition of analog vegetation states in the strict sense. We interpret a non-zero residual distance of $> 0.29$ between CDPs as a very high likelihood for a non-analog transient vegetation state (question 2), because it implies that even the equilibrium decade closest to the transient decade is still different from the transient decade (see Sect. S4 in the Supplement to get a detailed explanation of how we derived the 0.29 threshold value).

Contribution of individual key variables to the full Euclidean distance, i.e., the Euclidean distance calculated based on all nine state variables, was evaluated using a bootstrapping approach. Each variable was omitted and the reduced Euclidean distances based on the remaining eight key variables were calculated. The reduced distances were then set in relation to the full Euclidean distance to determine the percent deviation from the full distance caused by each variable: TS1

$$D_{x,y,t}^v = \frac{F_{x,y,t} - R_{x,y,t}^v}{F_{x,y,t}}. \tag{1}$$

Here, $F_{x,y,t}$ is the full Euclidean distance calculated using all nine state variables, at a given grid cell with coordinates $x$, $y$ for decade $t$, $R_{x,y,t}^v$ is the reduced Euclidean distance calculated based on eight state variables, omitting variable $v$ from the calculation, at a given grid cell with coordinates $x$, $y$ for decade $t$, and $D_{x,y,t}^v$ is the percent deviation from full Euclidean distance caused by omitting a given variable $v$ from distance calculation, at a given grid cell with coordinates $x$, $y$ for decade $t$.

Variables were then ranked for each grid cell and transient decade according to their percent deviation $D_{x,y,t}^v$ to determine the contribution of each variable to the full Euclidean distance $F_{x,y,t}$. The highest-contributing variable is termed "dominant variable" hereafter. Dominant variables were determined for SDPs as well as CDPs to answer question 3.

### 2.4.2 Biome classification

To assess which regions and vegetation formations in Africa are most resistant or most susceptible to future vegetation change (question 4), we aggregated vegetation in biomes using decadal averages of transient and equilibrium simulations following the scheme used in Scheiter et al. (2012) for all eight simulation scenarios. For definition of biome boundary criteria, see Table S1 in the Supplement.

To identify stable biome core areas for each of the eight scenarios, we identified grid cells with exactly one biome type in all 13 decades and created maps showing these areas. Desert core area was used for masking areas with very little vegetation to omit edge effects from such areas. Where grid cells took on more than one biome type in 13 decades, we counted the number of different biome types that occurred per grid cell, the number of changes between biome types per grid cell, and the ratio between biome types per grid cell and biome changes per grid cell. We created maps of these variables. Additionally, we defined each biome's area for all decades to determine changes in fractional cover over time for each scenario.

## 3 Results

### 3.1 Lags between equilibrium and transient simulations at continental scale

In simulations with fire, aboveground tree biomass in both equilibrium and transient scenarios was lower (Fig. 1a) and grass biomass was higher (Fig. 1b) than in no-fire scenarios. Seen in combination with the lower total tree cover in scenarios with fire (Fig. 1g), this indicates a more open landscape in the presence of fire. Average aboveground tree and grass biomass increased over time in all the scenarios. While tree biomass in transient scenarios was lower than in equilibrium scenarios, grass biomass in transient scenarios only dropped below levels expected based on equilibrium scenarios during the second half of the 21st century. Grass layer composition changed over time towards more $C_3$ and less $C_4$ grasses in all scenarios (Fig. 1c), with transient scenarios shifting to higher levels of $C_3$ grasses to a lesser degree than equilibrium scenarios. This indicates that the change is too slow to attain the levels of the equilibrium scenario.

While mean height of all trees combined (Fig. 1d) increased only slightly over time (in runs with fire) or remained more or less stable (in scenarios without fire), both maximum tree height (Fig. 1e) and number of tree individuals per unit area (Fig. 1f) increased over time, contributing to the simulated increase in tree biomass per unit area. Maximum tree height increased more strongly in equilibrium than transient simulations, with fire having very little effect due to tall trees not being affected by low- to medium-intensity fires in the aDGVM. The difference between transient and equilib-

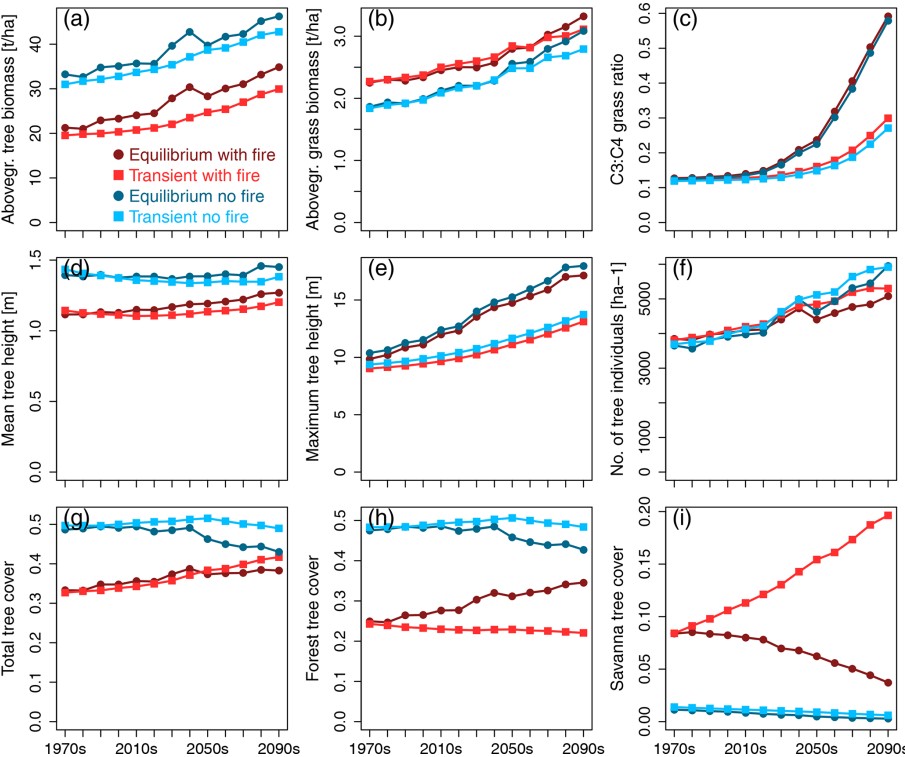

**Figure 1.** Time series of continental-scale spatial averages of variables for RCP8.5, calculated from decadal averages of grid cells.

rium states increased over time, showing that maximum tree height lags behind its equilibrium and lag size increases over time. Towards the end of the 21st century, tree numbers increased more strongly in no-fire simulations, and tree numbers were larger in transient than in equilibrium scenarios during the last decades.

Without the selection pressure exerted by fire, total tree cover (Fig. 1g) was essentially identical to forest tree cover (Fig. 1h) because savanna trees were largely absent in both equilibrium and transient simulations (Fig. 1i). While equilibrium simulations indicated more or less constant levels of total tree cover up to the year 2040, equilibrium tree cover declined after 2040 to approx. 42 % at the end of the century. In comparison, transient no-fire simulations suggested slightly rising total tree cover until 2050 followed by a slight decline to approx. 50 % cover towards the end of the century. Therefore, in the absence of fire, total transient tree cover increasingly deviated from total equilibrium tree cover during the second half of the century. The tree cover overshoot in no-fire transient simulations indicates that vegetation deviates from its equilibrium state.

The presence of fire fostered the existence of savanna trees in equilibrium and transient simulations (Fig. 1i). However, while the transient simulation showed an increase in savanna tree cover from approx. 8 % in the 1970s to approx. 20 % at the end of the century, equilibrium simulations showed a decline in savanna tree cover, with approximately half of the

original cover lost by the end of the century. While forest tree cover in transient simulations with fire decreased slightly from approx. 25 % to 21 % cover, it increased in equilibrium simulations and reached a value of approx. 34 % at the end of the century. In the presence of fire, both equilibrium and transient simulations showed a trend of increasing total tree cover over the course of the 21st century (Fig. 1g). However, while this increase was driven by an increase in forest tree cover that over-compensated for a simultaneous decline in savanna tree cover in equilibrium simulations, an increase in savanna tree cover caused the trend towards higher total tree cover in the transient simulation.

For the RCP4.5 climate scenario, the general patterns described for RCP8.5 were similar, but $C_3$ grasses did not become as prominent towards the end of the century as in RCP8.5 (see Fig. S1 for reference).

## 3.2 Similarity between same-decade partners

The Euclidean distance between SDPs averaged for Africa increased over time (Fig. 2). Fire consistently enlarged the distance between SDPs in comparison with the no-fire simulations (differences in spatial means between fire and no-fire partner scenarios were statistically significant at $p < 0.001$ based on $t$ tests and Kolmogorov–Smirnov tests) and led to the highest dissimilarity between SDPs in RCP8.5 towards the end of the century. RCP4.5 and RCP8.5 showed similar trajectories until the 2040s, but while the distance kept

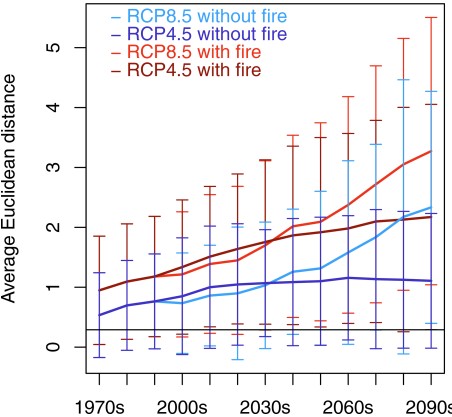

**Figure 2.** Continental-scale spatial average of Euclidean distance between same-decade partners (SDPs) for the four transient–equilibrium scenario pairings. Error bars represent standard deviation of spatial average in a given decade. The horizontal black line at 0.29 indicates the threshold value above which Euclidean distance is assumed to be significantly different from zero.

increasing towards the end of the century in RCP8.5, it leveled off in RCP4.5 with fire, and average distance remained approximately constant for RCP4.5 in the no-fire scenario. Spatial patterns of dissimilarity started to emerge during the first decades of the simulated period (Figs. 3, S2). In RCP8.5 with fire, maximum distance was found in the savanna areas south of the Congo basin and the Sahel zone during the 2010s (Fig. 3a), whereas no such pattern existed for the corresponding no-fire scenario (Fig. 3b). During the 2050s, the pattern of dissimilarity became more pronounced, and substantial distance between the transient and equilibrium scenarios was also observed in eastern and southeastern Africa (Fig. 3c). In the no-fire scenario, dissimilarity developed in eastern Africa and in western Angola (Fig. 3d). Towards the end of the century, distance between SDPs was substantial in most parts of Africa in RCP8.5 in both the fire and no-fire scenario. The largest distances were found in the Sahel, Ethiopia, and southern central Africa (Fig. 3e, f). The general spatial pattern observed in RCP8.5 was also found in RCP4.5 (Fig. S2) but was spatially less extensive and with overall lower distances between SDPs. Towards the end of the century, RCP4.5 had substantially lower distances than RC8.5, in particular in the scenario without fire.

## 3.3 Variable contributions to dissimilarity between SDPs

In RCP8.5 with fire, for ca. 28 % of African area savanna tree cover was the variable that had the largest influence on dissimilarity between SDPs in the 2010s (Fig. 4). Ranking of variables based on their impact on the full Euclidean distance between SDPs revealed that the variable with the strongest impact on average contributed ca. 40 % to the full Euclidean distance, whereas the variable with the second-strongest im-

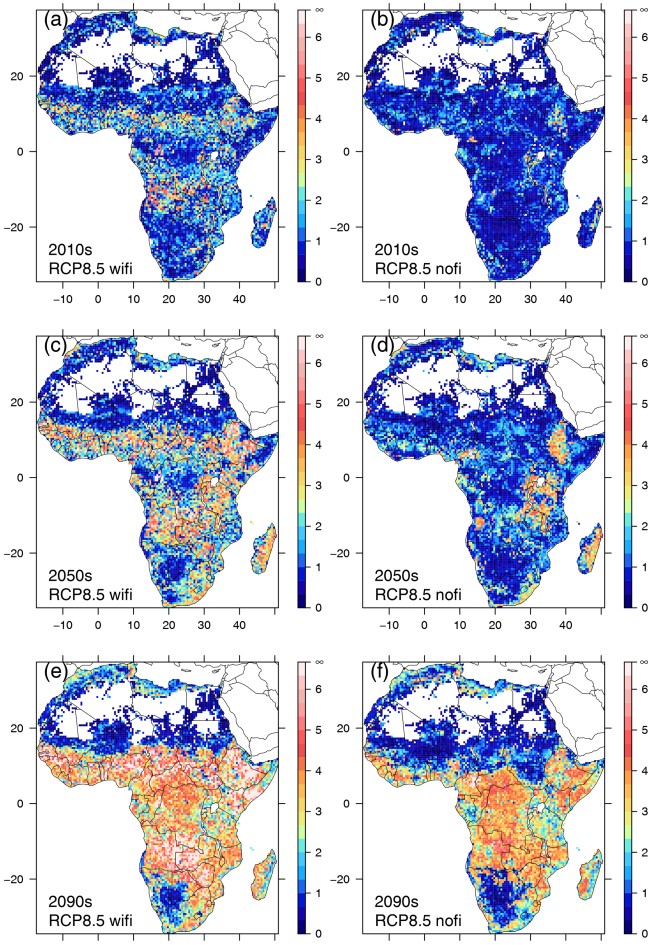

**Figure 3.** Spatial patterns of Euclidean distance between same-decade partners (SDPs) in RCP8.5 for three selected decades (2010–2019, 2050–2059, 2090–2099). Panels **(a)**, **(c)**, and **(e)** represent distances between SDPs in simulations including fire (wifi), **(b)**, **(d)**, and **(f)** show results from simulations excluding fire (nofi).

pact on average only contributed approx. 10 % (Fig. S3a). The strength of the impact varied between variables and was highest where mean tree height was identified as the most influential variable (ca. 65 % contribution) and lowest where forest tree cover was the most influential variable (ca. 18 % contribution). This general pattern was similar for all four scenarios (Fig. S3a, b, c, d). The area fraction where savanna tree cover had the largest contribution to dissimilarity increased towards mid-century and then slightly declined again towards the end of the century. Importance of average and maximum tree height was second and third after savanna tree cover in the 2010s, with the fraction of area where they dominated the Euclidean distance decreasing towards the end of the century. Remarkably, in RCP8.5 the area where the $C_3 : C_4$ grass ratio was the dominant variable increased towards the end of the century, which in this form was not found in either RCP4.5 scenario. In scenarios without fire,

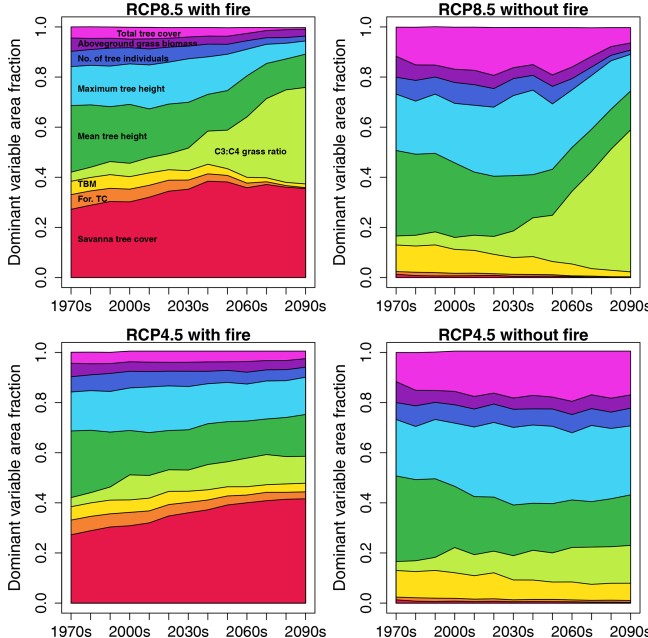

**Figure 4.** Fractions of African area where a given state variable is the dominant variable with respect to Euclidean distance between same-decade partners (SDPs), illustrated as time series stacks for the four scenario pairings between SDPs. Variable color coding is annotated in the panel for RCP8.5 with fire. The color coding for the variables is identical in all four panels.

savanna tree cover was very low and had less impact on Euclidean distance, where it was the dominant variable, while average and maximum tree height as well as total tree cover were more important. Maps of dominant variable distribution are shown in Figs. S4 and S5.

The percent deviance from the full Euclidean distance caused by the dominant variable, averaged across Africa and all decades, ranged between 40 % and 50 % (Fig. S6). Distinction of percent deviance according to dominant variables revealed some differences according to the identity of the dominant variable. Forest tree cover as the dominant variable caused a reduction of approx. 20 %, whereas mean tree height caused an approx. 60 % reduction from the full Euclidean distance where it was the dominant variable. This was fairly consistent for all four SDP combinations. The most pronounced difference between fire and no-fire scenarios was found with respect to savanna tree cover, which was largely irrelevant as a dominant variable in no-fire scenarios but also had less impact where it dominated than in the fire scenarios. For maps of percent deviance caused by the most influential variable, see Figs. S7 and S8.

### 3.4 Lag times between transient and closest-distance equilibrium vegetation states

The spatially averaged lag time between CDPs increased over time in all the scenarios (Fig. 5a). Until the 2030s, all the scenarios followed the same trajectory. After 2030, the scenarios with fire started to diverge from the scenarios without fire. At the end of the century, the spatially averaged lag time amounted to $5.0 \pm 3.5$ and $5.5 \pm 3.6$ decades for RCP8.5 and RCP4.5 with fire and $3.8 \pm 2.8$ and $4.4 \pm 3.1$ decades for RCP8.5 and RCP4.5 without fire, respectively.

While no clear spatial pattern in lag time existed in the 2010s (Fig. 6a), such a pattern emerged in the 2050s in RCP8.5 with fire (Fig. 6c) and had developed clearly during the last decade of the century (Fig. 6e). Lag times of 10 decades and more were found in the Sahel zone, eastern Angola, western Zambia, Zimbabwe, and the northeast of South Africa. In the no-fire RCP8.5 scenario, patterns were less clear and extreme lag times of a century or more were less abundant (Fig. S9). Patterns in RCP4.5 (Figs. S10, S11) were similar to those found in RCP8.5, but the boundaries between areas with large lag times and areas of moderate and intermediate lag times were more diffuse than in RCP8.5. In both RCP4.5 scenarios, lag times of 7–8 decades were more common at the end of the century in areas where lag times between 3 and 5 decades prevailed in RCP8.5.

### 3.5 Residual distance between closest-decade partners

Spatially averaged residual Euclidean distance between CDPs (Fig. 5b) was substantially smaller than for SDPs (Fig. 2) but nonetheless different from zero in all decades. The spatial variability of the size of the remaining Euclidean distance was high, especially towards the end of the century (see Fig. 6f), and the variables that were the main reason for the remaining Euclidean distance differed spatially across Africa (see Fig. 7 in combination with Fig. 5b to see the spatial fractions of variables that dominate the Euclidean distance at a given time). The non-zero distance between transient decades and closest equilibrium decades indicates that equilibrium states on average were still different from their transient partners. Residual distance was larger in both scenarios with fire compared to the respective no-fire partner scenarios and larger in RCP8.5 than RCP4.5 from mid-century onward. The closest agreement between CDPs was reached during the 2000s.

During the 2010s, residual distance between CDPs was below 1 in most regions of Africa in RCP8.5 with fire, except for areas adjacent to the north and south of the Congo basin, western Africa, and along the coast in southeastern Africa (Fig. 6b). In the no-fire scenario, residual distance was below 1 almost everywhere (Fig. S9b). By mid-century, the residual distance in the regions that already had elevated values in the 2010s had increased further, and additional areas of augmented distance had appeared in eastern Africa and the eastern parts of South Africa (Fig. 6d). In the no-fire scenario, residual distance was still low in most parts but started to increase in eastern Africa (Fig. S9d). At the end of the century, in RCP8.5 with fire substantial residual distance between CDPs existed in most parts of Africa, except for south-

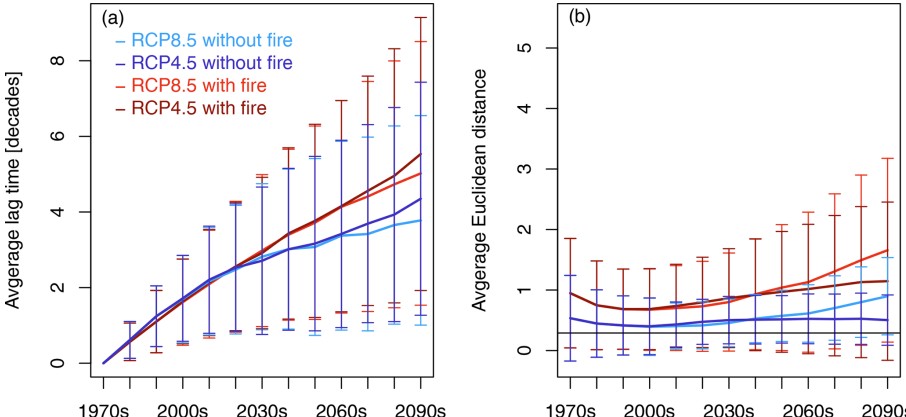

**Figure 5.** Continental-scale spatial average of lag time **(a)** and residual distance **(b)** between transient decade and most-similar equilibrium decade (closest-decade partners (CPDs) based on Euclidean distance) for the four scenario pairings between CDPs. Error bars represent standard deviation of spatial averages in a given decade. The horizontal black line at 0.29 in **(b)** indicates the threshold value above which Euclidean distance is assumed to be significantly different from zero. Lag time increases over time for all scenarios, and scenarios with fire start to diverge from scenarios without fire after 2030. Residual distances between CPDs are different from zero and indicate that transient vegetation states are not time-shifted trajectories of equilibrium vegetation states. To see which variables are the main drivers of the spatially averaged residual distance shown in **(b)**, please view **(b)** in comparison with Fig. 7.

western Africa, the central Congo basin, and the fringe areas of the Sahara (Fig. 6f), with maxima in eastern Africa and southern central Africa. In the no-fire scenario, residual distance had become more pronounced in eastern Africa since mid-century, and additional hotspot areas in Cameroon and Angola had developed (Fig. S9f).

The patterns for RCP4.5 were similar to those of RCP8.5 up to mid-century (Figs. S10, S11). However, residual distance towards the end of the century was considerably lower in both the fire and no-fire scenarios in RCP4.5.

### 3.6 Residual distance in relation to lag time

As shown in the preceding two sections, both lag time and residual distance on average increased over time and reached a maximum towards the end of the century. In all scenarios, residual distance tended to be lowest between CDPs that had a lag time of 4 decades (Fig. S12). Where CDPs exceeded lag times of 7 decades, residual distance increased with lag time in RCP8.5, especially in the scenario with fire. In RCP4.5, this increase was hardly visible (Fig. S12b) or absent (Fig. S12d).

### 3.7 Variable contributions to dissimilarity between CDPs

In most areas of Africa, a specific variable could be identified that dominated the Euclidean distance (Fig. S13). Savanna tree cover was the dominant variable explaining the distance between CDPs for 25 %–35 % of Africa's non-desert area in RCP8.5 with fire (Fig. 7). Mean tree height was the dominant variable for 26 % of Africa's non-desert area in the first decade in RCP8.5 (34 % in RCP4.5) and declined to 13 %

(17 %) towards the end of the century. Aboveground grass biomass was the dominant variable for 5 %–17 % of the area, with maximum extent reached in the 2010s. The area where the $C_3 : C_4$ grass ratio was the dominant variable increased towards the end of the century, where it reached a cover of approx. 21 % in RCP8.5 with fire. The overall pattern was similar in RCP4.5 with fire, with the exception that the $C_3 : C_4$ grass ratio never became as relevant as in RCP8.5. In scenarios without fire, savanna tree cover as a dominant variable for CDPs was negligible as this tree type was largely absent without fire. Consistent with the fire scenario, the RCP8.5 without fire showed an increase in area where the $C_3 : C_4$ grass ratio was the dominant variable towards the end of the century. For maps of dominant variable distribution, see Figs. S14 and S15.

The dominant variable for CDPs on average caused a 34 %–44 % deviation from the full residual distance (Fig. S16). Similar to SDPs, the impact caused by the dominant variable also depended on variable identity and for some variables varied between scenarios. In particular, savanna tree cover showed a difference between fire and no-fire scenarios, with its impact on full Euclidean distance being almost twice as high in fire than in no-fire scenarios. Where mean tree height was the dominant variable, it had the highest impact on residual distance but less than in SDPs and considerably less in RCP4.5 than RCP8.5. For spatial distribution of percent deviance caused by dominant variables, see Figs. S17 and S18.

### 3.8 Biome stability

Biome stability varied between scenarios (Fig. 8). Transient scenarios had larger stable areas across all decades than equi-

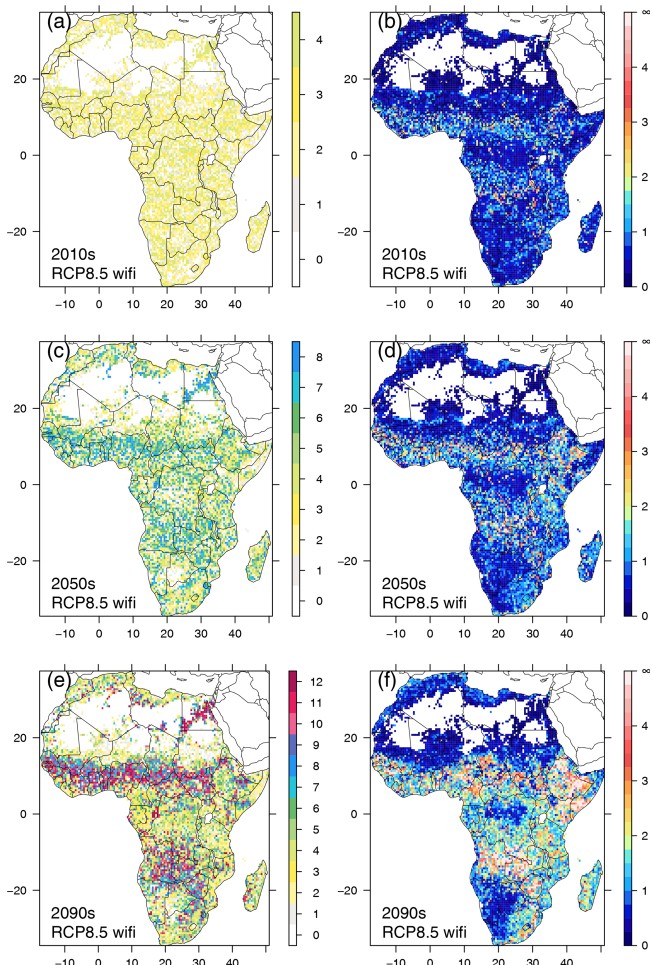

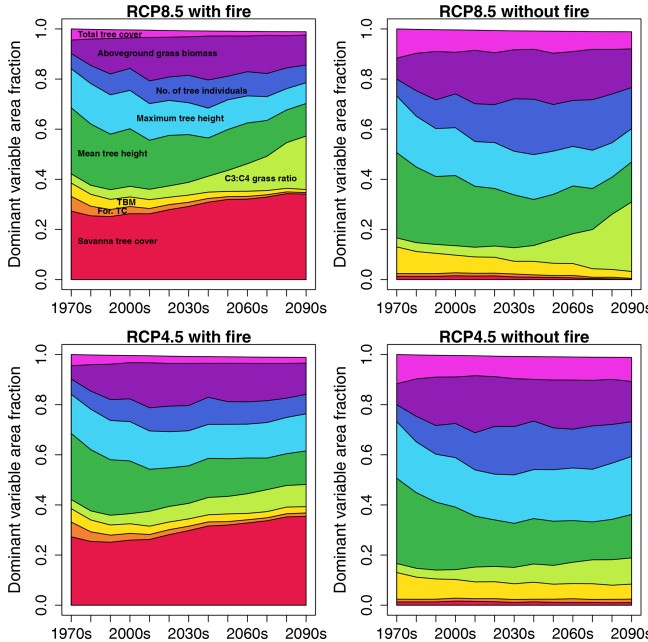

**Figure 7.** Fractions of African area where a given variable is the dominant variable defining residual distance between CDPs, illustrated as time series stacks for the four scenario pairings between CDPs. Variable color coding is annotated in the panel for RCP8.5 with fire. The color coding for the variables is identical in all four panels.

**Figure 6.** Spatial patterns of lag time (decades) between CDPs for RCP8.5 with fire **(a, c, e)** and residual Euclidean distance between CDPs **(b, d, f)**, for three selected decades (2010–2019, 2050–2059, 2090–2099).

scenarios revealed the highest number of biome changes and the most pronounced ratio between biome types and number of biome changes, indicating back-and-forth fluctuations between biome types. Consistent with the largest stable core sizes in no-fire transient scenarios, these also had the lowest numbers of biome types, biome changes, and the lowest ratios of biome types to biome changes.

librium scenarios, and no-fire scenarios had larger stable areas than the corresponding scenarios with fire. The largest stable areas were found in transient RCP4.5 without fire. Areas that experienced biome changes were located at the fringes of biome core areas, and fringe areas were consistently wider in equilibrium than in transient scenarios. Stable savanna core areas were absent in no-fire scenarios, where savanna core areas were replaced by woodland, and forest expanded into areas that were woodland or savanna in scenarios with fire (Fig. S19). $C_3$ grassland and $C_3$ savanna only emerged in small quantities in RCP8.5 with fire towards the end of the century. In the presence of fire, transient scenarios had more savanna areas than their equilibrium partners, which lost savanna area to woodland area towards the end of the century.

Where biome change occurred, the number of different biome types per grid cell was highest in the two equilibrium scenarios with fire (Figs. S20 and S21). Additionally, these

## 4  Discussion

Understanding time lags in the climate–vegetation system is important when trying to predict and evaluate vegetation dynamics, composition, structure, and associated ecosystem functions and services against the background of climate change. However, so far relatively few studies have focused on this topic. For example, Wu et al. (2015) and Chen and Wang (2020) studied time lag responses of vegetation growth to different climatic factors based on analysis of a time series of NDVI data. Papagiannopoulou et al. (2017) studied lagged vegetation anomalies caused by precedent precipitation based on multi-decadal satellite data. However, these studies were based on observational data, and therefore retrospectively, they focused on a small number of specific vegetation properties such as growth and NDVI and on lags occurring on timescales of months, seasons, or a few years. To our knowledge, our study is the first that models time lags

for future conditions, on a multi-decadal scale, focusing on the combined effects of different environmental drivers and a range of different key variables.

## 4.1  Key variable behavior and biome stability

Aboveground biomass increase was consistently observed across all scenarios for both trees and grasses (Figs. 1a and b, S1a and b). For trees, this biomass increase is due to an increase in maximum tree height (Figs. 1e, S1e) and in tree number (Figs. 1f, S1f) towards the end of the century and in scenarios with fire, also due to an increase in total tree cover (Figs. 1g, S1g). This persistent trend suggests that natural African vegetation may remain a carbon sink throughout the 21st century, although we have not specifically analyzed changes in carbon sink strength in this study. However, less biomass in transient than equilibrium scenarios towards the end of the century indicates carbon debt of ecosystems towards the atmosphere, which agrees with the findings of Scheiter et al. (2020). Hubau et al. (2020) found a stable carbon sink for Africa for the three decades up to 2015 and increased tree growth, consistent with the expected net effect of rising atmospheric $CO_2$, but predicted a long-term future decline in the African tropical forest sink. How the carbon balance of the African continent will develop is still subject to considerable uncertainty due to high interannual variability in emissions and involvement of a multitude of factors other than natural vegetation development. Human population development, land conversion, and biomass over-exploitation may severely impact Africa's potential as a future carbon sink (Williams et al., 2007; Brandt et al., 2017; Pelletier et al., 2018).

The simulated increase in biomass is likely linked to $CO_2$ fertilization effects. Woody encroachment, i.e., increase in woody vegetation cover, woody plant individuals, and woody biomass, is commonly observed in African savannas and often attributed to rising atmospheric $CO_2$ concentrations, although other factors such as water constraints, fire, and herbivory can confound the effect (Devinde et al., 2017; Case and Staver, 2017). As we did not conduct control simulations omitting $CO_2$ effects on vegetation, we cannot quantify how much of the biomass increase is due to rising $CO_2$ as opposed to other factors. However, when keeping climate constant in Scheiter et al. (2020) and varying $CO_2$, a positive effect of elevated $CO_2$ on carbon storage was observed. In two studies on biome change in South Asia (Kumar et al., 2020, in review) and Africa (Martens et al., 2020) we directly compared fixed $CO_2$ scenarios with scenarios following RCP8.5 and RCP4.5 climate and $CO_2$ trajectory. In these studies, we found that scenarios with fixed $CO_2$ experienced biomass decrease due to increased temperature and drought stress, whereas biomass increased in scenarios with elevated $CO_2$.

The degree to which $CO_2$ fertilization can (over-)compensate for vegetation die-back due to increased

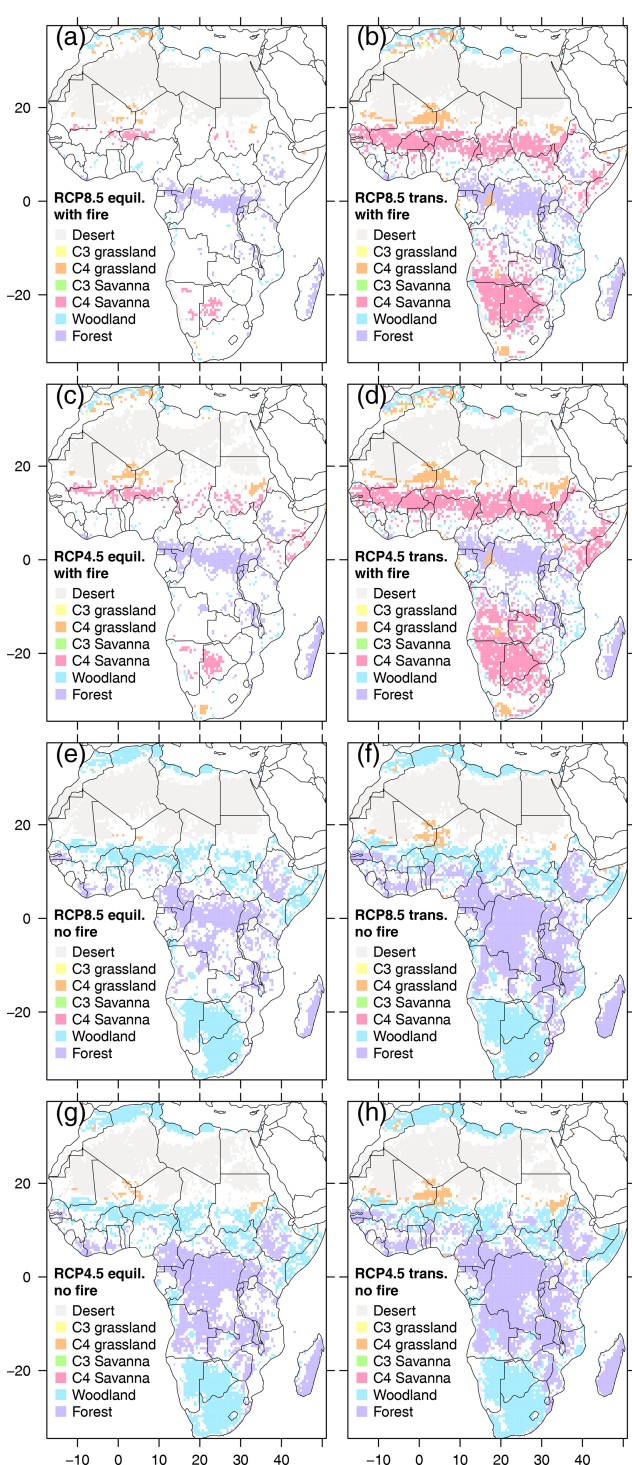

**Figure 8.** Areas with only one biome type in all 13 decades (i.e., biome core areas) shown for each of the 8 scenarios. Areas that experience one or more biome transitions are masked out (white areas). Transient scenarios are indicated by label "trans." (**b, d, f, h**), equilibrium scenarios by label "equil." (**a, c, e, g**).

temperature and water stress is limited (Jin et al., 2017; Xu et al., 2019; Jiang et al., 2020). Total tree cover decrease in our no-fire equilibrium simulations from mid-century onward (Figs. 1g, S1g) hints to such an upper limit. As conditions become drier towards the end of the century, even increased water use efficiency due to higher $CO_2$ becomes insufficient to sustain trees at the dry end of the gradient, and total tree cover decreases. Tree cover decline also occurs in the no-fire transient simulation but is less pronounced than in the equilibrium scenario. This indicates a tree cover surplus in the transient scenario that is meta-stable. In scenarios with fire, total tree cover is overall lower as fire reduces tree occurrence towards the dry range of the gradient; i.e., trees are already absent from sites that they can occupy in no-fire scenarios. The observed increase in tree biomass in no-fire scenarios is in contrast to the decline in tree cover and driven by tree number and maximum tree height; i.e., tree biomass increases because there are more trees and the maximum height of trees increases. The tree cover decline simulated by the aDGVM is likely yet an optimistic perspective. While water limitation effects on carbon assimilation and plant growth are captured, water stress mortality occurs only indirectly. Due to stomatal closure under water stress, the C balance of a simulated plant individual can become negative if respiratory costs exceed C gain, which increases the plant's probability of mortality. A more direct simulation of water stress-related effects, such as structural damage due to xylem cavitation, would likely further increase tree mortality and cover decline.

The pronounced increase in $C_3$ grasses towards the end of the century in RCP8.5 (Fig. 1c), but not in RCP4.5 (Fig. S1c), suggests that maximum $CO_2$ levels in RCP4.5 are not sufficient to enhance competitive performance of $C_3$ grasses such that they can coexist with or replace $C_4$ grasses in warm areas of Africa. This can be deduced from the fact that in RCP8.5 the $C_3 : C_4$ grass ratio is the dominant variable causing distance between SDPs towards the end of the century (Fig. 4). In RCP8.5, $C_3 : C_4$ grass ratio debt in transient simulations is the variable causing the largest difference between SDPs in many parts of Africa (Fig. S4c, f), but this is not the case in RCP4.5 (Fig. S5c, f). Even in areas where only little grass biomass exists, for example in the Congo basin, the difference in the $C_3 : C_4$ grass ratio between SDPs is larger than the differences caused by other key variables. This is because other key variables are comparably stable in these tropical forest areas. Although Euclidean distance is intermediate to high in this area (Fig. 3e, f), albeit lower than in savanna and woodland areas, up to 80 % or more of contribution to full Euclidean distance is explained by the dominant variable (Fig. S7e, f), i.e., by the $C_3 : C_4$ grass ratio. As amount of grass biomass is not considered in variable impact evaluation, the difference in the $C_3 : C_4$ grass ratio is the most prominent difference where other key variables are largely stable.

Aside from rainfall, fire plays a key role in landscape openness (Staver et al., 2011b), as indicated by lower levels of tree biomass (Figs. 1a, S1a), tree cover (Figs. 1g, S1g) and higher levels of grass biomass (Figs. 1b, S1b) in scenarios with fire as opposed to no-fire scenarios. Without fire pressure, savanna trees and savannas are largely absent and are replaced by woodland and forest (Figs. 1i, S1i, 8), which confirms findings that savannas and forests are alternative biome states differentiated by fire (Staver et al., 2011a). The bi-stability between woodland and savanna in the context of our study is the combined result of difference in tree type (dominant cover of forest or savanna trees) and tree cover in the presence or absence of fire. Savanna tree cover increases in transient but decreases in equilibrium simulations with fire (Figs. 1i, S1i), whereas total tree cover increases in both scenarios with fire (Figs. 1g, S1g). However, this total tree cover increase is driven by forest trees in equilibrium and by savanna trees in transient simulations. Where we see the final stage of succession as represented by the aDGVM in the equilibrium scenario, what we see in the transient scenario is a snapshot of a system in motion. The increase in savanna tree cover in the transient scenarios can thus be interpreted as intermediary disequilibrium stage that already indicates transition towards more tree cover but has not yet moved to the next successional stage that would be replacement of savanna trees with forest trees. Anthropogenic fire management may therefore have considerable effects on vegetation state and carbon sequestration of African ecosystems. For example, Scheiter et al. (2015) showed that different fire return intervals and early vs. late dry season management fires influence biomass and other state variables of simulated biomes. Scheiter and Savadogo (2016) showed that management can slow or accelerate tipping point behavior and hence the magnitude of vegetation lags. Targeted fire reduction could help to shift African vegetation towards higher woody cover and biomass and therefore increase the size of the African carbon sink. This would, however, lead to the loss of unique ecosystem types and their associated biodiversity and ecosystem functions. In particular grasslands and savannas are threatened by targeted fire reductions, because fire plays a pivotal role in the dynamics of these ecosystem types. Conservation management therefore has to balance trade-offs between carbon storage vs. ecosystem conservation when evaluating the role of fire as a management tool in African ecosystems.

Areas of biome stability are more extensive in transient than equilibrium scenarios (Fig. 8), and where biome change occurs a higher number of biome types is simulated and a larger number of biome changes occurs in equilibrium scenarios (Figs. S20, S21). Areas that are stable in transient but not in equilibrium scenarios can be interpreted as meta-stable legacy states. The recognition of such meta-stable states has important implications for conservation. Conservation of meta-stable states will require extra effort as the system may ultimately move towards a stable state. Areas of biome stability are also more extensive in no-fire than in fire scenarios,

indicating the role of fire in keeping vegetation in dynamic disequilibrium. More forests and woodlands in no-fire equilibrium scenarios strongly support the notion that in our simulations a large part of the savannas exists due to disturbance, with fire keeping vegetation in fluctuation between a mix of intermediary successional stages (Meyer et al., 2009).

### 4.2 Dissimilarity between same-decade partner scenarios

Euclidean distance between SDPs increased over time (Figs. 2, 3, S2), which was more pronounced in fire than in no-fire scenarios. Such an increase in distance can be an indication of time lags in vegetation dynamics as well as of non-analog vegetation states. Whether the former or the latter or a combination of both causes the observed dissimilarity cannot be discerned based only on SDP comparison. A difference between RCP4.5 and RCP8.5 was found for the second half of the century, with dissimilarity in RCP4.5 only moderately rising but further increasing in RCP8.5, where $CO_2$ keeps rising and climate continues to change.

The vegetation formations most at risk are savannas and woodlands due to their meta-stability. They show highest dissimilarity between transient and equilibrium state (compare Figs. 3, S4 and 8 for RCP8.5, and Figs. S2, S5 and 8 for RCP4.5), which implies that they are farthest from their equilibrium stage and therefore most at threat to experience change even after a stabilization of climate and $CO_2$ concentrations. Savannas are disturbance-driven systems that are subject to continuous fluctuations caused by abiotic and biotic disturbances. Due to these non-equilibrium processes that characterize savannas, they are non-equilibrium systems that fluctuate around a mean state classifying them as savannas (Gillson, 2004). If climate change deflects savannas to a degree where this mean state changes from savanna to woodland or forest, they additionally may become a disequilibrium vegetation formation, i.e., a vegetation formation that does not correspond to the new equilibrium state demanded by the forcing regime. They will then be a remnant of a foregone forcing system due to a relaxation time that exceeds the time it took the forcing to change. It is likely that this disequilibrium state will entail leading-edge as well as trailing-edge dynamics. Leading-edge effects include lags due to migration and local population built-up and succession, whereas trailing edge effects are caused by delayed local extinctions and slow losses of ecosystem structural components (Svenning and Sandel, 2013). Our results indicate that savannas are particularly sensitive to future change in environmental drivers, because in fire-scenarios, differences in savanna tree cover were the dominant driver for SDP dissimilarity for 25 % to 40 % of African non-desert area (Fig. 4). Our results therefore suggest that savannas are likely to become disequilibrium vegetation formations and therefore will need special focus in conservation management.

### 4.3 Dissimilarity between closest-decade partners

Increasing lag times between CDPs (Fig. 5a) and increasing dissimilarity of SDPs over time (Fig. 2) are a sign that environmental drivers change faster than vegetation can adapt. This agrees with findings of Jezkova and Wiens (2016) that rates of change in climatic niches in plant and animal populations are much slower than projected climate change, posing a threat in particular to tropical species. Extreme lag times can therefore indicate areas where environmental drivers change at a particularly high rate, where transient vegetation is in a meta-stable state, and where future tipping of vegetation into alternative stable states is likely. Conversely, areas with low lag times can either indicate low rate of change in environmental drivers at the regional scale or indicate vegetation that is particularly resistant to changing environmental conditions. In both cases, small vegetation changes are sufficient to stay close to the anticipated equilibrium state, either because change in environmental drivers is weak and does not require much change in vegetation or because equilibrium vegetation is stable across a wide range of environmental drivers. Lag size can therefore be explained by combined evaluation of change in environmental drivers and vegetation resistance.

Combining information on vegetation lag time with residual distance between CDPs (Fig. 5a and b) reveals that transient vegetation states are likely non-analog to any simulated equilibrium state. If transient vegetation states were on a time-shifted trajectory of equilibrium vegetation states, residual distances between CDPs should be approx. zero. This is not the case in our comparison of CDPs (Fig. 5b), where spatially averaged residual distance ranges between 0.5 and 1.5 depending on scenario and decade. Spatially explicit evaluation of simulations with fire showed that areas of particularly high residual distance (Figs. 6b, d, f, S10b, d, f) were mostly located in savanna and woodland areas to the north and south of the Congo basin, in eastern Africa and eastern South Africa. Fire caused more pronounced residual distances between CDPs than found in no-fire scenarios, where areas of pronounced dissimilarity only started to emerge towards the end of the century (Figs. S9b, d, f, S11b, d, f). This is a strong indication that disturbances can help to keep vegetation in meta-stable intermediary successional states (Dantas et al., 2016). Comparison of residual distance patterns (Fig. 6b, d, f) with lag time patterns (Fig. 6a, c, e) reveals a connection between areas of pronounced residual distance and long lag times. This implies that although a closest equilibrium partner was found, this partner not only has a vegetation state that corresponds to past environmental conditions but also is a poor match for the transient vegetation state. We deduce from this that the corresponding simulated transient vegetation states are composite non-analog states that cannot be described by any vegetation state achievable under equilibrium conditions.

Residual distance between CDPs is dominated by different key variables depending on location (Figs. S14, S15). In scenarios including fire, differences in savanna tree cover dominated dissimilarity between CDPs in roughly a quarter to a third of African non-desert area (Fig. 7), which supports the notion that savanna and woodland areas are bi-stable states (Higgins and Scheiter, 2012; Staal et al., 2016) and therefore prone to tipping point behavior in the future (Gillson, 2015). $CO_2$ concentrations anticipated under RCP8.5 for the second half of the century are predicted to cause shifts from $C_4$ to $C_3$ dominance in the grass layer in extensive areas of Africa (Figs. 1c, S5e, f). The threshold $CO_2$ levels at which such a shift in dominance occurs is also influenced by growing-season temperature and water availability and additionally influenced by non-climatic factors such as fire, herbivore preferences and light availability (Ehleringer, 2005). Whether these shifts will be realized also depends on the availability of a $C_3$ grass species pool in these areas. Environmental niche suitability alone not necessarily implies realization of niche occupancy when target organisms (in this case $C_3$ grasses) are absent, e.g., due to migrational lags and local dispersal limits (Dexiecuo et al., 2012).

Non-analog transient vegetation states emerge due to differing response times of key processes and state variables, leading to cumulative lagged responses that act on different biodiversity components, including individuals, populations, species and communities (Essl et al., 2015b). In Scheiter et al. (2020), we showed time series of different state variables at a savanna study site in South Africa that illustrated the temporal sequence of process and state variable responses from leaf level to population level. While ecophysiological responses such as increased photosynthesis happen very quickly, population-level responses are slower and respond sequentially on different timescales. This implies that vegetation in transient state is subject to multiple lags, i.e., at any given time different key variables have different individual lag times. These multiple lags make it impossible to approximate transient vegetation states through equilibrium states, resulting in composite non-analog states.

The finding that future transient vegetation states deviate from any equilibrium state has implications for conservation management. Conservation managers need to decide on target ecosystem states, and whether preservation of contemporary ecosystem states will be feasible and sustainable in the future. Awareness of meta-stable vegetation states should influence decisions on suitable intervention measures, and help decide to what extent these need to be applied (Gillson, 2015). In this context, our study can help to identify those vegetation types and areas that are most prone to change and tipping point behavior in the face of future climate change and therefore need particular focus. We found that savannas and woodlands, or more generally speaking those systems where disturbance regime is important, are especially likely to exhibit multi-lags and meta-stability. This is because disturbances such as fire or herbivory cause cyclical successional resets that keep systems from converging to late-successional states (Meyer et al., 2007), and therefore can exacerbate climate-driven lags and meta-stability (Scheiter et al., 2020). Accordingly, climate-mediated changes in disturbance regime also need consideration in conservation management, e.g., changes in fire frequency, intensity, or timing of occurrence (Battisti et al., 2016).

## 4.4 Opportunities and limitations of this study

Field surveys and remote sensing data provide valuable information on vegetation status. However, they are usually limited with respect to the time span they can cover, and they are subject to a trade-off between high spatial or high temporal resolution, as well as between high spatial resolution and spatial extent. In addition, observations are also confined to the past or present. Without reference base, it is hard to determine whether an observed vegetation state is in equilibrium with environmental forcing, time-lagged, or non-analog. Dynamic vegetation modeling can overcome these constraints. Moreover, the influence of specific driver variables can be studied in isolation, e.g., the effect of elevated $CO_2$ can be studied by keeping climate constant (Scheiter et al., 2020). Dynamic vegetation modeling also offers the possibility to generate equilibrium vegetation states by enforcing constant or detrended drivers and allowing the model to reach equilibrium under these conditions. These simulated equilibrium vegetation states can then be used as controlled reference base for simulated transient vegetation states but also to assess the status of observed vegetation. Enforcement of vegetation equilibrium, projection of future vegetation states, and the possibility for isolated factorial analysis of specific drivers using vegetation models therefore provides a unique opportunity to address knowledge gaps that cannot be filled by observation data.

A limitation of the approach presented in this study is that climate data availability for RCP8.5 and RCP4.5 determined the starting point (in our case the 1970s) for both equilibrium and transient vegetation simulations. This holds the implicit assumption that transient and equilibrium vegetation state were similar at the starting point. Moreover, the conceptual setup implies that simulated lag times cannot exceed the number of decades between the 1970s and the decade of interest. Therefore, simulated distance and lag times between the historic decades and present can be underestimated and need to be seen with caution, as observed vegetation states in Africa during the 1970s were very likely not in equilibrium with environmental conditions of the 1970s. Hence, where lag time equals number of simulated decades, the lag time and associated Euclidean distances represent a lower limit estimate. Consequently, simulated lag times and Euclidean distances in some cases may be underestimated due to the limitation caused by the need to start simulations at the beginning of the climate data set. We are, however, confident that the general message of the simulation experiment, i.e., that

transient vegetation states are non-analog to equilibrium vegetation states, and lag behind forcing, is nonetheless valid.

We only conducted a limited number of equilibrium simulation runs to establish equilibrium vegetation states as reference basis. The decadal-scale discretization was chosen because 13 simulation runs per scenario were determined as technically feasible while also ensuring variability in input climate data. However, discretization could imply that residual distance between CDPs may be overestimated if the best equilibrium match to a transient vegetation state was located between two equilibrium scenarios. However, given the clear dominance of specific key variables for residual distance between CDPs, we deem it unlikely that discretization is responsible for overestimates of residual distances large enough to falsely assume non-analog state for a given transient vegetation state. Moreover, an analysis of lag times conducted for single variables revealed a large range of variability in lag times between variables for a given transient decade, especially in the second half of the century (not shown). This is a clear sign of multi-lags that should be unrelated to discretization and therefore points to true non-analog transient vegetation states.

Fire in the aDGVM does not account for explicit occurrence of ignitions but has heuristically been calibrated such that the ignition rates and resulting fires agree well with observed fire patterns and frequency (Scheiter and Higgins, 2009). Where occurrence of ignitions may change in the future, e.g., due to changes in fire management or occurrence of lightning strikes due to climate change, the aDGVM may therefore miss such changes in ignition patterns. However, given that the majority of African ecosystems are currently not ignition-limited and therefore climate and landscape connectivity combined with human fire management strategies are the main limiting factors on fire occurrence (Archibald et al., 2012, and references therein), the simulated amount of fire is driven by the other two components of the fire triangle (fuel load and quality, fire weather conditions, e.g., fuel moisture). As fire intensity and spread in the aDGVM are linked to fuel moisture, fuel biomass and tree cover (increasing tree cover reduces fire occurrence), simulated fire regimes in the future do change in response to climate and vegetation change in a non-ignition-limited system even if changes in ignition patterns are not directly captured themselves. We therefore estimate that our main findings regarding the role of fire in relation to vegetation patterns and lags would not change substantially with explicit representation of ignitions.

Due to the large number of simulation runs required for this study (56 runs in total), we only used downscaled climate output data from one Earth system model (ESM). The results might therefore differ slightly when using climate output data from other ESMs. However, results from another study recently conducted with the aDGVM for Africa using CCAM-downscaled projections from six different ESMs showed that the choice of ESM had the smallest effect on simulation outcome (Martens et al., 2020). Variation between all 24 ensemble members in that study was mainly explained by the $CO_2$ scenario, followed by interactions between $CO_2$ and RCP scenarios, while the type of ESM had only minor influence. The biomass values simulated with the downscaled MPI-ESM climatology in that study were slightly below the mean of the six ensemble members, indicating a tendency towards slightly more-than-average temperature increase and MAP decrease. This agrees with the slightly above-average equilibrium climate sensitivity (ECS) value of 3.6 for MPI-ESM-LR (ensemble mean: $3.2 \pm 1.3$, in Table 9.5 of Flato et al., 2013). Given the low impact of the ESM scenario on the results and the fact that the downscaled MPI-ESM climatology used in this study lies close to the ensemble mean of different ESMs, we are confident that our results are representative although only output from one ESM was used.

All presented simulations were conducted offline, i.e., without direct coupling between vegetation and climate. We expect that lag times, bi-stability, and non-linear tipping behavior between different vegetation states could be even more pronounced in an online-coupling experiment, because stability is likely enhanced by feedback mechanisms that foster it. For example, tropical rain forests transfer large quantities of water vapor to the atmosphere and locally create clouds and precipitation sustaining their existence even if regional-scale precipitation patterns without such feedbacks showed decreasing trends (see, e.g., Staal et al., 2018). In line with Zhu and Zeng (2014), we expect that albedo effects, canopy transpiration and evaporation, and temperature effects mitigated by vegetation could alter local to regional climate, in turn feeding back onto vegetation dynamics. In semi-arid areas, such feedbacks can decide which one of several possible equilibrium states will be realized, e.g., whether grasslands or deserts will be realized as alternative stable states (Zeng et al., 2004). However, even fully coupled ESMs may be unable to predict how future feedbacks between vegetation and climate will shape terrestrial vegetation state, as shown by Bathiany et al. (2014) in the context of future Sahel greening trends simulated by three different ESMs with dynamic vegetation coupling.

## 5 Conclusions

Our results show that simulated transient vegetation states increasingly deviate from equilibrium vegetation states in both RCP scenarios, and that during the second half of the 21st century this deviation is more pronounced in RCP8.5 than RCP4.5. Fire additionally increased Euclidean distance between SDPs due to its restraining effects on vegetation succession. Individual key variables such as woody cover, grass and tree biomass, and tree height differed between transient and equilibrium scenarios, and for many regions variables that dominated Euclidean distance between transient and equilibrium partner scenarios could be clearly identified. Tra-

jectories of transient vegetation change did not follow a "virtual trajectory" of equilibrium states; i.e., they are not time-shifted trajectories of equilibrium vegetation states but composite non-analog states caused by multiple lags with respect to vegetation processes and composition. Lag times between transient and most similar equilibrium vegetation states increased over time and to a degree were found to agree with spatial patterns of maximum residual Euclidean distance between CDPs. Extremely long lag times can be indicative of high rates of change in environmental drivers, of non-analog transient vegetation states, and of meta-stability and risk of future tipping points. Lag times toward the end of the century were most pronounced in savanna and woodland areas north and south of the Congo basin, the Sahel zone, eastern Africa, and eastern South Africa, with savanna tree cover frequently being the main driver of transient–equilibrium dissimilarities in these regions. Our results indicate that savanna ecosystems will be most at risk for shifts towards alternative stable states and therefore need a strong focus in nature conservation management.

*Code and data availability.* The aDGVM code used to produce the results presented in this publication is available here: https://doi.org/10.5281/zenodo.4108449 (Pfeiffer et al., 2020a). The decadally averaged model output data analyzed in this study as well as the scripts used to conduct data analysis and to create the figures shown in the paper and its Supplement are available at https://doi.org/10.17605/OSF.IO/64MGK (Pfeiffer et al., 2020b).

*Supplement.* The supplement related to this article is available online at: https://doi.org/10.5194/bg-17-1-2020-supplement.

*Author contributions.* MP and SS conceived the study. MP designed and conducted the analysis of simulation results and led the writing of this article. SS conducted the simulations for this study and contributed to the analysis of simulation results. DK and CM provided valuable support for the implementation of this study. All the authors contributed to the writing of this article.

*Competing interests.* The authors declare that they have no conflict of interest.

*Acknowledgements.* Mirjam Pfeiffer thanks the German Federal Ministry of Education and Research (BMBF) for funding (SPACES II initiative, "SALLnet" project). Simon Scheiter and Dushyant Kumar thank the Deutsche Forschungsgemeinschaft (DFG) for funding. Carola Martens thanks the German Federal Ministry of Education and Research (BMBF) for funding (SPACES II initiative, "EMSAfrica").

*Financial support.* This research has been supported by the Bundesministerium für Bildung und Forschung (grant nos. 01LL1802B and 01LL1801B) and the Deutsche Forschungsgemeinschaft (grant no. SCHE 1719/2-1).

The publication of this article was funded by the Open Access Fund of the Leibniz Association.

*Review statement.* This paper was edited by Martin De Kauwe and reviewed by two anonymous referees.

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

**Remarks from the typesetter**

TS1    To ensure a transparent review process, I will have to ask the editor for approval of this correction. Could you please provide a short explanation for this correction or let me know if the one from the previous proof-reading can be forwarded to the editor? Thank you very much in advance for your help.