# Peer review of "Climate change will cause non-analogue vegetation states in Africa and commit vegetation to long-term change"

_Biogeosciences, 2020_

## Referee Comment (RC1) · Anonymous Referee #1 · 3 Aug 2020

Thank you for inviting me to review paper: "Climate change will cause non-analogue vegetation states in Africa and commit vegetation to long-term change" by Pfeiffer et al.

The central premise of the Abstract is that transients in the vegetation response imply that the land surface does not merely behave as a set of time-evolving equilibrium states as the background climate changes. Instead, inertia implies alternative vegetation features might exist and that are only possible in a transient situation. Maybe not surprisingly, these are most notable under RCP8.5 ("business-as-usual" situation).

Maybe be even more explicit why the expression "non-analogue" is used throughout.

This suggestion is because often “analogue” can refer to simply anything that is different to states that have only been observed, (either in the recent past or possibly paleo-records). Here “non-analogue” implies non-pseudo equilibrium – so states that are not equilibrium either past, contemporary or projected under climate change. Possibly an alternative term could be something like “novel transient”.

The second line in the Abstract “This implies that vegetation is committed to future changes once environmental drivers stabilise” is important, and it might be good to re-iterate that towards the end. Something in general language might be useful e.g. “conservation managers. . .. . ...should be aware that observed vegetation may continue to change substantially, even if climate drivers are held fixed”.

The Introduction is good, and it recognises that the way vegetation sees differences between equilibrium and transient responses. The Introduction makes it clear that equilibrium-transient differences can be in both the multiple elements of the climatological drivers, and in the lags of the land surface itself (affecting its structure and composition). I also like that the aims of the paper are made very clear with the bullet points 1,2,3 at the end of the Introduction.

However, like many readers, I also looked at the conclusions before reading the main bulk of the paper. Notable is that the conclusions state: “ . . .shift towards alternative stable states”. So in other words, the transient time-history of vegetation evolution may impact on different final equilibrium states, even for the same equilibrium forcings. The vegetation of Africa has always been speculated as capable of that (i.e. “multi-stable vegetation coverage”; there are many references to this). It feels as if this should be listed as an extra point 4 in the Introduction, given it is discussed in this manuscript.

It is interesting that the effects of fire can have such a substantial impact on the magnitude of lags behind any equilibrium state. Does the paper hint at targeted fire reductions i.e. by deliberate human intervention could be useful in some circumstances?

The most interesting summary diagram in my view is Figure 5. It very cleverly shows

an overall lag of vegetation from equilibrium in the left-hand panel, while the right-hand panel calculates a residual term which captures the “non-analogue” distance from any past equilibrium solution. As these days, people often extract diagrams and captions from papers to put in to powerpoint talks, would it help to expand slightly the caption to this diagram.

I also have a small request concerning Figure 5. The units of the left-hand panel are intuitive, as time lags (decades). The right-hand panel is Euclidean distance, based around the nine state variables (p9) contributing to Equation (1) (p10). I cannot think of an answer to this, but it would be good if there was some sort of physical or biological units/quantities associated with the right-hand panel of Figure 5. OK, maybe readers need to then look at Figure 7, which shows which biome is most different when compared to the nearest equilibrium decade. Hence write the manuscript to encourage the reader to view figure 5 and Figure 7 simultaneously?

It would be good to see an expanded version of “Opportunities and limitations of this study”. First, if I have understood the paper correctly, then only one overall forcing Earth System Model (ESM) is used - as then disaggregated by CCAM. That model is the MPI-ESM ESM. The author should state where this model sits in terms of its equilibrium climate sensitivity (ECS). Is it a fast or slow warming model – or ideally towards the middle of any distribution? The ECS numbers are available in the 5th IPCC report. I realise this is technically challenging, given the need to disaggregate via CCAM, but future work could include more ESMs and from both the CMIP5 and CMIP6 ensemble.

A second point for the “limitations” section is it feels to me as if there needs to be much more confidence in the fire model. In particular, the Methods section states “ignitions are based on a random sequence”. That randomness might have to change in time, if for instance, it includes lightning strikes, the frequency of which are likely to vary under global warming. It is noted that every diagram in the paper has both with fire and without fire findings presented equally. Future analysis, with a well-established

and tested fire model, should give emphasis to the simulations with fire, as they are the more process-complete simulations.

A third point for the "limitations" section is that all the analysis presented is offline. The authors might like to speculate whether they think more multiple-stable states exist if the vegetation is coupled to an atmospheric model, thus allowing for feedbacks. There is a very long literature on this, some of which might be good to cite here. See for instance, Zeng et al. "Multiple equilibrium states and the abrupt transitions in a dynamical system of soil water interacting with vegetation" and the many references in that paper.

Broadly I like this paper and I think with some minor adjustments, it is suitable for publication. I am very happy to see any revised manuscript version.

Small additional things

The Abstract feels a bit too technical in places e.g. use of word "Euclidean".

Figure 1 (and maybe similar elsewhere). The fonts of the labels and the legends appear very small. One possibility to make more space – at least in the vertical direction – could be to only mark the "x"-axis labels under panels g,h,i.

Figure 3 – the colourbar levels look slightly odd. It feels to me as if they would be neater if simply 0.0, 1.0, 2.0, 3.0, . . ..

Please check through again in general the diagrams. For instance, I realise it is obvious, but the convention in Figure 4 would be "biomes types are as annotated in panel a. The colours used are common between all four panels".

Figure 8, with the small font used in the map annotations, it took me some time to realise that the "t" and "e" mentioned in the caption to Figure 8 was added to the end of those annotations. Hence e.g. "RCP8_5e". Please improve the presentation of this diagram, along with the caption and the annotations.

---

## Author Comment (AC1) · 3 Sep 2020

**Author responses to comments of anonymous referee 1**
**Responses are highlighted in bold font.**

Thank you for inviting me to review paper: "Climate change will cause non-analogue vegetation states in Africa and commit vegetation to long-term change" by Pfeiffer et al.

**Thank you for taking the time and making the effort to read and evaluate our**

[Figure]

**manuscript.**

The central premise of the Abstract is that transients in the vegetation response imply that the land surface does not merely behave as a set of time-evolving equilibrium states as the background climate changes. Instead, inertia implies alternative vegetation features might exist and that are only possible in a transient situation. Maybe not surprisingly, these are most notable under RCP8.5 ("business-as-usual" situation). Maybe be even more explicit why the expression "non-analogue" is used throughout. This suggestion is because often "analogue" can refer to simply anything that is different to states that have only been observed, (either in the recent past or possibly paleo-records). Here "non-analogue" implies non-pseudo equilibrium – so states that are not equilibrium either past, contemporary or projected under climate change. Possibly an alternative term could be something like "novel transient".

**Thank you for pointing out the difficulties of the term "non-analogue". We are aware that "non-analogue" is often used in the context of comparison between palaeo-vegetation states and present or future vegetation states that have not been found in this form in the past. However, what we refer to is the comparison between (hypothetical) pseudo-equilibrium states and the composite transient vegetation states that cannot be represented by any of the pseudo-equilibrium states. We found it difficult to find a term that would describe this discrepancy in an appropriate way and therefore decided to use the term "non-analogue". As you suggest, we will add a more concrete definition of how we define "non-analogue" in the context of the study (i.e., in the sense of not having an equivalent equilibrium state) in the introduction section to make it as clear as possible what we mean.**

The second line in the Abstract "This implies that vegetation is committed to future

changes once environmental drivers stabilise" is important, and it might be good to re-iterate that towards the end. Something in general language might be useful e.g. "conservation managers. . .. . ...should be aware that observed vegetation may continue to change substantially, even if climate drivers are held fixed".

**We will pick up your suggestion and add a corresponding sentence in that sense towards the end of the abstract.**

The Introduction is good, and it recognises that the way vegetation sees differences between equilibrium and transient responses. The Introduction makes it clear that equilibrium-transient differences can be in both the multiple elements of the climatological drivers, and in the lags of the land surface itself (affecting its structure and composition). I also like that the aims of the paper are made very clear with the bullet points 1,2,3 at the end of the Introduction.

**Thank you, we are glad that the introduction convincingly transferred the message to you that we wanted to convey.**

However, like many readers, I also looked at the conclusions before reading the main bulk of the paper. Notable is that the conclusions state: " . . .shift towards alternative stable states". So in other words, the transient time-history of vegetation evolution may impact on different final equilibrium states, even for the same equilibrium forcings. The vegetation of Africa has always been speculated as capable of that (i.e. "multi-stable vegetation coverage"; there are many references to this). It feels as if this should be listed as an extra point 4 in the Introduction, given it is discussed in this manuscript.

**We will include a mention of the possibility that shifts towards alternative stable states may be affecting African ecosystems as a consequence of climate**

**change in the introduction, as you suggest, and include references to studies that focused on this topic. We know that multi-stability of ecosystem states, in particular in connection with Africa's savanna ecosystems, has been studied and proposed previously by a variety of authors (e.g., Staal et al., 2016, Li et al., 2019; Pausas & Bond, 2020). Therefore, we decided to not focus on this topic again in our study. Ass we are not specifically investigating muli-stable states in this study, it is not a direct aim/working hypothesis and we therefore will not put it with the bullet points at the end of the introduction, but keep it within the more general part of the introduction.**

It is interesting that the effects of fire can have such a substantial impact on the magnitude of lags behind any equilibrium state. Does the paper hint at targeted fire re-ductions i.e. by deliberate human intervention could be useful in some circumstances?

**We are not considering fire management effects in this study, but have done so in previous studies with aDGVM. In Scheiter et al. (2015), we showed how different fire return intervals and early vs. late dry season management fires influence biomass and other state variables. In Scheiter & Savadogo (2016) we showed that management can slow down or accelerate tipping point behavior and hence the magnitude of lags. The effect of fire on vegetation state is ecosystem-specific and strongly depends on the management goals. Without fire, the majority of open and semi-open ecosystems in Africa are simulated to display higher woody cover and biomass. Targeted fire reduction therefore could help to increase the size of the African carbon sink. This would, however, come at the cost of losing unique ecosystem types and their associated biodiversity and ecosystem functions. In particular grasslands and savanna ecosystems are threatened by targeted fire reductions as fire plays a pivotal role in the dynamics of these ecosystem types. We will add a brief statement highlighting**

**these conflicting fire management targets (trade-off between carbon storage vs. ecosystem conservation) in section 4.1 of the discussion.**

The most interesting summary diagram in my view is Figure 5. It very cleverly shows an overall lag of vegetation from equilibrium in the left-hand panel, while the right-hand panel calculates a residual term which captures the "non-analogue" distance from any past equilibrium solution. As these days, people often extract diagrams and captions from papers to put in to powerpoint talks, would it help to expand slightly the caption to this diagram.

**Thank you for the comment. We will provide a more detailed figure caption in the revised version of the manuscript in order to make the figure self-explanatory without having to rely on the manuscript's main text. We suggest the following expanded figure caption:** *Continental-scale spatial average of lag time (panel a) and residual distance (panel b) between transient decade and most-similar equilibrium decade (closest-decade partners (CDPs) based on Euclidean distance), for the four scenario pairings between CDPs. Error bars represent standard deviation of spatial averages in a given decade. Lag time increases over time for all scenarios, and scenarios with fire start to diverge from scenarios without fire after 2030. Residual distances between CDPs are different from zero and indicate that transient vegetation states are not time-shifted trajectories of equilibrium vegetation states.*

I also have a small request concerning Figure 5. The units of the left-hand panel are intuitive, as time lags (decades). The right-hand panel is Euclidean distance, based around the nine state variables (p9) contributing to Equation (1) (p10). I cannot think of an answer to this, but it would be good if there was some sort of physical or biological units/quantities associated with the right-hand panel of Figure 5. OK, maybe

readers need to then look at Figure 7, which shows which biome is most different when compared to the nearest equilibrium decade. Hence write the manuscript to encourage the reader to view figure 5 and Figure 7 simultaneously?

**We are well aware that Euclidean distance is a general measure that integrates over a variety of possible causes. Based on the distance alone, it is indeed not possible to discern the major cause that underlies the difference. In addition, due to the normalization of variables used to derive the Euclidean distance, the distance itself becomes unit-less, i.e., abstract.Therefore, your idea to more explicitly point out the connection between Euclidean distance in Fig. 5b and Fig. 7 that shows the fractions of variables that dominate the Euclidean distance at a given time is quite helpful to make the integrated distance shown in Fig. 5b more tangible for the reader. We will add a sentence to results section 3.7 and to the caption of Fig. 5 to encourage readers to view both figures conjointly in order to compare the average size of the distance at a given time with the respective fractional variable contributions.**

It would be good to see an expanded version of "Opportunities and limitations of this study". First, if I have understood the paper correctly, then only one overall forcing Earth System Model (ESM) is used - as then disaggregated by CCAM. That model is the MPI-ESM ESM. The author should state where this model sits in terms of its equilibrium climate sensitivity (ECS). Is it a fast or slow warming model – or ideally towards the middle of any distribution? The ECS numbers are available in the 5th IPCC report. I realise this is technically challenging, given the need to disaggregate via CCAM, but future work could include more ESMs and from both the CMIP5 and CMIP6 ensemble.

**Thank you for pointing this limitation out, we will add it to the limitations discus-**

**sion section (4.3). We are currently in the process of publishing a parallel study
where we used aDGVM in combination with an ensemble of CCAM-downscaled
projections of climate change under RCP4.5 and RCP8.5 from 6 different ESMs
to test the influence of increasing $CO_2$ concentration on carbon storage and
to evaluate uncertainties regarding future biome change.In this study, we
quantified the effect size of the explanatory variables ($CO_2$ scenario, i.e., fixed
vs. elevated $CO_2$, RCP scenario and GCM) and their two-way interactions on
the variability of the dependent variables (aboveground vegetation carbon and
water use efficiency) between 2000-2019 and 2080-2099 using $\omega^2$ metric. The $\omega^2$
metric indicated that variation in total carbon between all 24 ensemble members
was mainly explained by the $CO_2$ scenario, followed by interaction effects of
$CO_2$ and RCP scenarios. The choice of GCM had the smallest effect on the
simulation outcome. The biomass values simulated with MPI-ESM were slightly
below the ensemble mean of the 6 ensemble members, which could indicate
a tendency towards slightly more-than-average temperature increase and MAP
decrease. This result agrees with the slightly-above average ECS value of 3.6
for MPI-ESM-LR (ensemble mean: 3.2 $\pm$ 1.3, Tab. 9.5, IPCC2014). With this, the
MPI-ESM is none of the models that lie on the edge of the range and therefore
should provide reasonable input climatology.Due to the necessity to conduct 13
equilibrium simulation runs per RCP and fire scenario in this study, in addition
to the 4 transient runs (i.e., 56 simulation runs total for one ESM), we refrained
from using more than one ESM's output in order to keep computational time
within feasible limits.**

A second point for the "limitations" section is it feels to me as if there needs to be much
more confidence in the fire model. In particular, the Methods section states "ignitions
are based on a random sequence". That randomness might have to change in time,
if for instance, it includes lightning strikes, the frequency of which are likely to vary
under global warming. It is noted that every diagram in the paper has both with fire

and without fire findings presented equally. Future analysis, with a well-established and tested fire model, should give emphasis to the simulations with fire, as they are the more process-complete simulations.

**We agree that fire is a complex disturbance regime that depends on many influencing factors and is associated with various uncertainties. For Africa, it is estimated that the majority of ecosystems are currently not ignition-limited, i.e., ignition rates are more than sufficient to burn the available fuel, so climate and landscape connectivity combined with human fire management strategies are the main limiting factors on fire occurrence (Archibald et al., 2012, and references therein). Although the current implementation of fire in aDGVM does not account for explicit ignitions, it has heuristically been calibrated such that the ignition rates and resulting fires agree well with observed fire patterns and fire frequency (Scheiter & Higgins 2009). Therefore, the calibrated ignitions in aDGVM at least for Africa should not be limiting, even if currently not modeled explicitly. This implies that the simulated amount of fire is driven by the other two components of the fire triangle, i.e., fuel load and quality and fire weather conditions (i.e., fuel moisture). As fire intensity and spread in aDGVM are linked to fuel moisture, fuel biomass and tree cover (increasing tree cover reduces fire spread), fire regimes thereby change in response to climate change and vegetation change. Based on past personal experience from developing a more complex process-based fire model (Pfeiffer et al., 2013), I can say that such a detailed implementation of fire-related processes not necessarily improves the accuracy of fire representation due the increasing number of parameters that need to be estimated and defined, which increases uncertainty. We will add a section in the discussion where we elaborate on the points mentioned here.**

A third point for the "limitations" section is that all the analysis presented is offline. The

authors might like to speculate whether they think more multiple-stable states exist if the vegetation is coupled to an atmospheric model, thus allowing for feedbacks. There is a very long literature on this, some of which might be good to cite here. See for instance, Zeng et al. "Multiple equilibrium states and the abrupt transitions in a dynamical system of soil water interacting with vegetation" and the many references in that paper.

**It is hard to speculate how an online coupling between aDGVM and an ESM would influence simulated vegetation dynamics due to the non-linearity of feedback mechanisms the spatially differentiated nature of such effects that will vary between different types of ecosystems. We can therefore only provide rather speculative answers to that question. The work of Zeng et al. (2004) suggests that multiple equilibrium states are possible in semi-arid areas, with grasslands vs. desert being alternative stable states. They also suggest that the range of parameter space over which these equilibria can coexist may be increased by positive feedbacks of evapotranspiration on precipitation (e.g., Wang & Eltahir, 2000). Zhu & Zeng (2014) evaluated the difference between offline and online simulations, but vegetation in their simulations was prescribed and therefore could not respond to climate change. In line with Zhu & Zeng (2014), we would expect that in particular albedo effects, canopy transpiration and evaporation, and temperature effects mitigated by vegetation could alter local to regional climate, in turn feeding back on vegetation dynamics. Where such two-way feedback mechanisms between vegetation and climate exist, we would expect that lag times, bi-stability and non-linear tipping behavior between different vegetation states could be even more pronounced, because stability is likely enhanced by feedback mechanisms that foster such stability. For example, tropical forests that transfer large quantities of water vapor to the atmosphere via transpiration locally create clouds and precipitation that sustain the existence of such forests even if regional-scale precipitation patterns without**

**such feedbacks showed decreasing trends (see, e.g., Staal et al., 2018). In that sense, such forests foster climatic conditions that sustain their existence. However, even fully coupled ESMs may be unable to consistently predict how future feedbacks between vegetation and climate will shape terrestrial vegetation state, as shown by Bathiany et al. (2014) in the context of future Sahel greening trends simulated by three different ESMs with dynamic vegetation. We will add a section on this topic to the "limitations" discussion section.**

Broadly I like this paper and I think with some minor adjustments, it is suitable for publication. I am very happy to see any revised manuscript version.

**Thank you.**

Small additional things

The Abstract feels a bit too technical in places e.g. use of word "Euclidean".

**We will rephrase the sentence with the first occurrence of the term "Euclidean distance" to make it more clear that this is used as a measure of dissimilarity between vegetation states. i.e., the sentence "Euclidean distance between simulated transient and equilibrium vegetation states based on selected variables was used to determine lag times and similarity of vegetation states" will be rephrased as follows: "We determined lag times and dissimilarity between simulated and transient vegetation states based on the combined difference of 9 selected state variables using Euclidean distance as a measure for that difference." In line 15/16, the term "Euclidean distance" will be replaced by "dissimilarity".**

Figure 1 (and maybe similar elsewhere). The fonts of the labels and the legends appear very small. One possibility to make more space – at least in the vertical direction – could be to only mark the "x"-axis labels under panels g,h,i.

**Thank you for pointing this out. Given the possibility to zoom in on figures in digital form, we apparently did not pay enough attention to this. We will re-plot this figure and increase font sizes as much as possible to ensure that labels and legends are more easily legible, in particular on multi-panel figures.**

Figure 3 – the colourbar levels look slightly odd. It feels to me as if they would be neater if simply 0.0, 1.0, 2.0, 3.0, . . ..

**We will re-plot the figures in the main text and the supplementary material with a colorbar that has more even breaks.**

Please check through again in general the diagrams. For instance, I realise it is obvious, but the convention in Figure 4 would be "biomes types are as annotated in panel a. The colours used are common between all four panels".

**Thank you for pointing this out, we will change the figure captions according to your suggestion.**

Figure 8, with the small font used in the map annotations, it took me some time to realise that the "t" and "e" mentioned in the caption to Figure 8 was added to the end of those annotations. Hence e.g. "RCP8_5e". Please improve the presentation of this diagram, along with the caption and the annotations.

**We will increase the font size of the legend as far as possible and highlight the caption of the legend in bold to make it easier to read, and change the figure caption to point out which of the panels are representing transient and which ones are representing equilibrium scenarios.**

References

Archibald, S., Staver, A. C., and Levin, S. A.: Evolution of human-driven fire regimes in Africa, Proceedings of the National Academy of Sciences, 17:109, 847–852, https://doi.org/10.1073/pnas.1118648109, 2012.

Bathiany, S., Claussen, M., and Brovkin, V.: CO2-induced Sahel greening in three CMIP5 Earth System Models, Journal of Climate, 27, 7163–7184, https://doi.org/10.1175/JCLI-D-13-00528.1, 2014.

Li, Q., Staver, A. C., E., W., Levin, S. A.: Spatial feedbacks and the dynamics of savanna and forest, Theoretical Ecology, 12, 237–262, https://doi.org/10.1007/s12080-019-0428-1, 2019.

Pausas, J. G., and Bond, W. J.: Alternative biome states in terrestrial ecosystems, Trends in Plant Science, 25, 250–263, https://doi.org/10.1016/j.tplants.2019.11.003, 2020.

Pfeiffer, M., Spessa, A., and Kaplan, J. O.: A model for global biomass burning in preindustrial time: LPJ-LMfire (v1.0), Geosci. Model Dev. 6, 643–685, https://doi.org/10.5194/gmd-6-643-2013, 2013.

Scheiter, S., Higgins, S. I., Beringer, J. and Hutley, L. B.: Climate change and long-term fire management impacts on Australian savannas, New Phytologist, 205, 1211–1226, https://doi.org/10.1111/nph.13130, 2015.

Scheiter, S., and Savadogo, P.: Ecosystem management can mitigate vegetation shifts induced by climate change in West Africa, Ecological Modelling, 332, 19-27,

https://doi.org/10.1016/j.ecolmodel.2016.03.022, 2016.

Staal, A., Dekker, S. C., Xu, C., van Nes, E. H.: Bistability, spatial interaction, and the distribution of tropical forests and savannas, Ecosystems, 19, 1080–1091, https://doi.org/10.1007/s10021-016-0011-1, 2016.

Staal, A., Tuinenburg, O.A., Bosmans, J.H.C. et al.: Forest-rainfall cascades buffer against drought across the Amazon, Nature Clim Change, 8, 539–543, https://doi.org/10.1038, 2018.

Wang, G., and E. A. B. Eltahir: Biosphere-atmosphere interactions over West Africa, 2. Multiple climate equilibria, Q. J. R. Meteorol. Soc., 126, 1261–1280, https://doi.org/10.1002/qj.49712656504, 2000.

Wuyts, B., Champneys, A. R., House, J. I.: Amazonian forest-savanna bistability and human impact, Nature Communications, 8, 15519, https://doi.org/10.1038/ncomms15519, 2019.

Zeng, X., Shen, S. P., Zeng, X., and Dickinson, R.: Multiple equilibrium states and the abrupt transitions in a dynamic system of soil water interacting with vegetation, Geophysical Research Letters 31, L05501, https://doi.org/0.1029/2003GL018910, 2004.

Zhu Jia-Wen & Zeng Xiao-Dong (2014) Comparison of the Influence of Interannual Vegetation Variability between Offline and Online Simulations, Atmospheric and Oceanic Science Letters, 7:5, 453–457, https://doi.org/10.3878/j.issn.1674-2834.14.0031

---

## Referee Comment (RC2) · Anonymous Referee #2 · 11 Sep 2020

The authors present a theoretical study on possible vegetation changes in Africa for two scenarios of global warming and climate change. They use the sophisticated and well documented aDGVM, a dynamic (but not global) vegetation model that has been developed specifically for grass-tree interaction in tropical ecosystems. The authors convincingly demonstrate that in a global warming scenario, the vegetation composition in Africa will likely change and increasingly deviate from its equilibrium composition, i.e., its composition that is attained, if vegetation would instantaneously follow the changing climate. In this sense, the transient future vegetation state in Africa is supposed to move into 'non-analogue states'.

In conclusion, this is a well written, interesting study. The method is clearly outlined. The results are thoroughly and convincingly discussed. The topic is highly relevant. I am happy to recommend its publication in Biogeosciences in its present form subject to a few small, editorial changes.

Minor items:

Line 234: Fire 'consistently' enlarges... ok, but what about statistical significance? I assume the scatter is just too large to talk about statistical significance. This is more a comment, which the authors might consider, not a critical remark.

Lines 249 to 252: I had to read these sentences at least twice to fully understand their content. Which variables refer to which percentage? Perhaps a slight re-arrangement of the sentence starting with 28% will cure the problem. It slightly enhances understanding, if the authors more specifically refer to Fig. S3a, instead of Fig. S3 (and if the 'Fig. S3a' were put in closed brackets).

Line 363: What are these unpublished studies by the co-authors (Kumar and Martens)? Grey literature, PhD theses, to be submitted, or just personal communication?

Figures: The figure captions should be self-explaining as much as possible. Therefore, please, explain the acronyms (SDP in Fig.2, 3, 4 and CDP in Fig. 5, 6, 7 and the figures in the Supplement)

---

## Author Comment (AC2) · 21 Sep 2020

Author responses to comments of anonymous referee 2.
**Responses are highlighted in bold font.**

The authors present a theoretical study on possible vegetation changes in Africa for two scenarios of global warming and climate change. They use the sophisticated and well documented aDGVM, a dynamic (but not global) vegetation model that has been developed specifically for grass-tree interaction in tropical ecosystems. The authors convincingly demonstrate that in a global warming scenario, the vegetation

composition in Africa will likely change and increasingly deviate from its equilibrium composition, i.e., its composition that is attained, if vegetation would instantaneously follow the changing climate. In this sense, the transient future vegetation state in Africa is supposed to move into 'non-analogue states'. In conclusion, this is a well written, interesting study. The method is clearly outlined. The results are thoroughly and convincingly discussed. The topic is highly relevant. I am happy to recommend its publication in Biogeosciences in its present form subject to a few small, editorial changes.

**Thank you for taking the time and making the effort to read and evaluate our manuscript. We are happy that you found it interesting and worthwhile for publication in Biogeosciences.**

Minor items: Line 234: Fire 'consistently' enlarges. . . ok, but what about statistical significance? I assume the scatter is just too large to talk about statistical significance. This is more a comment, which the authors might consider, not a critical remark.

**We intentionally wrote "consistently" instead of "significantly" because we did not test for statistical significance when aggregating data for Figure 2. The scatter is indeed very large, as indicated by the plotted standard deviations of the spatial means in Fig. 2. This wide scatter is a consequence of the distinct spatial patterning of Euclidean distance emerging over time that can be seen in Fig. 3 and Fig. S2. It is likely that the difference in Euclidean distance between fire and no-fire scenarios is significant for specific regions where fire strongly drives vegetation dynamics, and this then reflects in the consistent difference of the continental-scale mean, which in itself may not be significant. We will add a brief explanation on this topic when presenting the results of Fig. 2 and 3, and can conduct a test for significant difference of continental-scale mean values**

**between fire and no-fire scenarios.**

Lines 249 to 252: I had to read these sentences at least twice to fully understand their content. Which variables refer to which percentage? Perhaps a slight re-arrangement of the sentence starting with 28% will cure the problem. It slightly enhances under-standing, if the authors more specifically refer to Fig. S3a, instead of Fig. S3 (and if the 'Fig. S3a' were put in closed brackets).

**Thank you for pointing out your difficulties with these sentences, as well as the formatting issue with the brackets. We will rephrase these sentences to communicate our point more clearly. As an alternative way of phrasing, we suggest the following:** *"In RCP8.5" with fire, for 28% of vegetated African area savanna tree cover was the variable that had the largest influence on dissimilarity between SDPs in the 2010s (Fig. 4). Ranking of variables based on their impact on the full Euclidean distance between SDPs revealed that the variable with the strongest impact in average contributed ca. 40% to the full Euclidean distance, whereas the variable with the second-strongest impact in average only contributed approx. 10% (Fig. S3a). The strength of impact varied between variables and was highest where mean tree height was identified as most influential variable (ca. 65% contribution), and lowest where forest tree cover was the most influential variable (ca. 18% contribution). This general pattern was similar for all four scenarios (Fig. S3a, b, c, d)."*

Line 363: What are these unpublished studies by the co-authors (Kumar and Martens)? Grey literature, PhD theses, to be submitted, or just personal communication?

**The study of Kumar et al. is meanwhile published as a discussion article (Kumar,**

**D., Pfeiffer, M., Gaillard, C., Langan, L., and Scheiter, S.: Climate change and elevated CO2 favor forest over savanna under different future scenarios in South Asia, Biogeosciences Discuss., https://doi.org/10.5194/bg-2020-169, in review, 2020.), and the study of Martens et al. is currently under review with Global Change Biology. We will update the references accordingly.**

Figures: The figure captions should be self-explaining as much as possible. Therefore, please, explain the acronyms (SDP in Fig.2, 3, 4 and CDP in Fig. 5, 6, 7 and the figures in the Supplement)

**We will update the figure captions according to your suggestion to make them self-explanatory.**

---

## Author Response (AR1)

**Comments from Reviewers and Authors' response: Climate change** will cause non-analogue vegetation states in Africa and commit vegetation to long-term change**

Mirjam Pfeiffer1, Dushyant Kumar1, Carola Martens1,2, and Simon Scheiter1

1Senckenberg Biodiversity and Climate Research Centre (BiK-F), Senckenberganlage 25, 60438 Frankfurt am Main, Germany
2Institute of Physical Geography, Goethe University Frankfurt am Main, Altenhoeferallee 1, 60438 Frankfurt am Main, Germany

Correspondence: Mirjam Pfeiffer (mirjam.pfeiffer@senckenberg.de)

**1 Author responses to comments of anonymous referee #1**

**Responses are highlighted in bold font.**

Thank you for inviting me to review paper: "Climate change will cause non-analogue vegetation states in Africa and commit vegetation to long-term change" by Pfeiffer et al.

**5 Thank you for taking the time and making the effort to read and evaluate our manuscript.**

The central premise of the Abstract is that transients in the vegetation response imply that the land surface does not merely behave as a set of time-evolving equilibrium states as the background climate changes. Instead, inertia implies alternative vegetation features might exist and that are only possible in a transient situation. Maybe not surprisingly, these are most no-

- 10 table under RCP8.5 ("business-as-usual" situation). Maybe be even more explicit why the expression "non-analogue" is used throughout. This suggestion is because often "analogue" can refer to simply anything that is different to states that have only been observed, (either in the recent past or possibly paleo-records). Here "non-analogue" implies non-pseudo equilibrium – so states that are not equilibrium either past, contemporary or projected under climate change. Possibly an alternative term could be something like "novel transient".
- 15 Thank you for pointing out the difficulties of the term "non-analogue". We are aware that "non-analogue" is often used in the context of comparison between palaeo-vegetation states and present or future vegetation states that have not been found in this form in the past. However, what we refer to is the comparison between (hypothetical) pseudoequilibrium states and the composite transient vegetation states that cannot be represented by any of the pseudoequilibrium states. We found it difficult to find a term that would describe this discrepancy in an appropriate way and
- 20 therefore decided to use the term "non-analogue". Following your suggestion, we have added a more concrete definition of how we define "non-analogue" in the context of the study (i.e., in the sense of not having an equivalent equilibrium state) in the introduction section to make it as clear as possible what we mean (p.3 lines 1-4 and p.3 lines 16-18).

The second line in the Abstract "This implies that vegetation is committed to future changes once environmental drivers

25 stabilise" is important, and it might be good to re-iterate that towards the end. Something in general language might be useful e.g. "conservation managers. . . . . ...should be aware that observed vegetation may continue to change substantially, even if climate drivers are held fixed".

We followed your suggestion and have added a corresponding sentence at the end of the abstract to highlight the implications for conservation management (p.1 lines 49-52).

30

35

The Introduction is good, and it recognises that the way vegetation sees differences between equilibrium and transient responses. The Introduction makes it clear that equilibrium-transient differences can be in both the multiple elements of the climatological drivers, and in the lags of the land surface itself (affecting its structure and composition). I also like that the aims of the paper are made very clear with the bullet points 1,2,3 at the end of the Introduction.

Thank you, we are glad that the introduction convincingly transferred the message to you that we wanted to convey.

However, like many readers, I also looked at the conclusions before reading the main bulk of the paper. Notable is that the conclusions state: "...shift towards alternative stable states". So in other words, the transient time-history of vegetation evolution may impact on different final equilibrium states, even for the same equilibrium forcings. The vegetation of Africa has always been speculated as capable of that (i.e. "multi-stable vegetation coverage"; there are many references to this). It feels as if this should be listed as an extra point 4 in the Introduction, given it is discussed in this manuscript.

We now briefly discuss the possibility that shifts towards alternative stable states may be affecting African ecosystems as a consequence of climate change in the introduction, as you suggest, and have included additional references to studies that focused on this topic (p.2 lines 41-47). We know that multi-stability of ecosystem states, in particular in connection with Africa's savanna ecosystems, has been studied and proposed previously by a variety of authors (e.g., Staal et al., 2016; Li et al., 2019; Pausas & Bond, 2020). Therefore, we decided to not focus on this topic again in our study. As we are not specifically investigating mulistable states in this study, it is not a direct aim/working hypothesis and we therefore did not put it with the bullet points at the end of the introduction, but have kept it within the more general part of the introduction.

It is interesting that the effects of fire can have such a substantial impact on the magnitude of lags behind any equilibrium state. Does the paper hint at targeted fire reductions i.e. by deliberate human intervention could be useful in some circumstances?

55 We are not considering fire management effects in this study, but have done so in previous studies with aDGVM. In Scheiter et al. (2015), we showed how different fire return intervals and early vs. late dry season management fires influence biomass and other state variables. In Scheiter & Savadogo (2016) we showed that management can slow down or accelerate tipping point behavior and hence the magnitude of lags. The effect of fire on vegetation state is ecosystemspecific and strongly depends on the management goals. Without fire, the majority of open and semi-open ecosystems

in Africa are simulated to display higher woody cover and biomass. Targeted fire reduction therefore could help to 60 increase the size of the African carbon sink. This would, however, come at the cost of losing unique ecosystem types and their associated biodiversity and ecosystem functions. In particular grasslands and savanna ecosystems are threatened by targeted fire reductions as fire plays a pivotal role in the dynamics of these ecosystem types. We have added a brief discussion highlighting these conflicting fire management targets (trade-off between carbon storage vs. ecosystem con-

65 servation) in section 4.1 of the discussion (p.15 lines 14-32).

The most interesting summary diagram in my view is Figure 5. It very cleverly shows an overall lag of vegetation from equilibrium in the left-hand panel, while the right-hand panel calculates a residual term which captures the "non-analogue" distance from any past equilibrium solution. As these days, people often extract diagrams and captions from papers to put in to powerpoint talks, would it help to expand slightly the caption to this diagram.

Thank you for the comment. We now provide a more detailed figure caption in the revised version of the manuscript in order to make the figure self-explanatory without having to rely on the manuscript's main text (see Fig. 5 and its new caption on page 10 of the mark-up version of the revised manuscript).

- I also have a small request concerning Figure 5. The units of the left-hand panel are intuitive, as time lags (decades). The 75 right-hand panel is Euclidean distance, based around the nine state variables (p9) contributing to Equation (1) (p10). I cannot think of an answer to this, but it would be good if there was some sort of physical or biological units/quantities associated with the right-hand panel of Figure 5. OK, maybe readers need to then look at Figure 7, which shows which biome is most different when compared to the nearest equilibrium decade. Hence write the manuscript to encourage the reader to view figure 5 and
- 80 Figure 7 simultaneously?

We are well aware that Euclidean distance is a general measure that integrates over a variety of possible causes. Based on the distance alone, it is indeed not possible to discern the major cause that underlies the difference. In addition, due to the normalization of variables used to derive the Euclidean distance, the distance itself becomes unit-less, i.e., abstract. Therefore, your idea to more explicitly point out the connection between Euclidean distance in Fig. 5b and

Fig. 7 that shows the fractions of variables that dominate the Euclidean distance at a given time is quite helpful to make 85 the integrated distance shown in Fig. 5b more tangible for the reader. We have added a sentence to results section 3.5 (p.10 lines 4-13) and to the caption of Fig. 5 (see Fig. 5 and its new caption on page 10 of the mark-up version of the revised manuscript) to encourage readers to view both figures conjointly in order to compare the average size of the distance at a given time with the respective fractional variable contributions.

90

70

It would be good to see an expanded version of "Opportunities and limitations of this study". First, if I have understood the paper correctly, then only one overall forcing Earth System Model (ESM) is used - as then disaggregated by CCAM. That model is the MPI-ESM ESM. The author should state where this model sits in terms of its equilibrium climate sensitivity (ECS). Is it a fast or slow warming model – or ideally towards the middle of any distribution? The ECS numbers are available

95 in the 5th IPCC report. I realise this is technically challenging, given the need to disaggregate via CCAM, but future work could include more ESMs and from both the CMIP5 and CMIP6 ensemble.

100

105

Thank you for pointing this limitation out. We have added it to the limitations discussion section (p.17 lines 195-109, p.18 lines 1-10), where we now discuss the climatology of MPI-ESM in comparison to the climatology simulated by other ESMs, and we provide additional information on the sensitivity of aDGVM simulation results to the used climatology in comparison to the sensitivity to other factors (RCP scenario, CO2 forcing).

A second point for the "limitations" section is it feels to me as if there needs to be much more confidence in the fire model. In particular, the Methods section states "ignitions are based on a random sequence". That randomness might have to change in time, if for instance, it includes lightning strikes, the frequency of which are likely to vary under global warming. It is noted that every diagram in the paper has both with fire and without fire findings presented equally. Future analysis, with a

well-established and tested fire model, should give emphasis to the simulations with fire, as they are the more process-complete simulations.

We agree that fire is a complex disturbance regime that depends on many influencing factors and is associated with various uncertainties. For Africa, it is estimated that the majority of ecosystems are currently not ignition-limited, i.e.,

- 110 ignition rates are more than sufficient to burn the available fuel, so climate and landscape connectivity combined with human fire management strategies are the main limiting factors on fire occurrence (Archibald et al., 2012, and references therein). Although the current implementation of fire in aDGVM does not account for explicit ignitions, it has heuristically been calibrated such that the ignition rates and resulting fires agree well with observed fire patterns and fire frequency (Scheiter and Higgins 2009). Therefore, the calibrated ignitions in aDGVM at least for Africa should
- 115 not be limiting, even if currently not modeled explicitly. This implies that the simulated amount of fire is driven by the other two components of the fire triangle, i.e., fuel load and quality, and fire weather conditions (i.e., fuel moisture). As fire intensity and spread in aDGVM are linked to fuel moisture, fuel biomass, and tree cover (increasing tree cover reduces fire spread), fire regimes thereby change in response to climate change and vegetation change. Based on past personal experience from developing a more complex process-based fire model (Pfeiffer et al., 2013), I can say that such
- 120 a detailed representation of fire-related processes not necessarily improves the accuracy of fire representation due the increasing number of parameters that need to be estimated and defined, which increases uncertainty. We have added a paragraph in the discussion where we elaborate on the points mentioned here (p.17 lines 70-94).

A third point for the "limitations" section is that all the analysis presented is offline. The authors might like to speculate whether they think more multiple-stable states exist if the vegetation is coupled to an atmospheric model, thus allowing for feedbacks. There is a very long literature on this, some of which might be good to cite here. See for instance, Zeng et al. "Multiple equilibrium states and the abrupt transitions in a dynamical system of soil water interacting with vegetation" and the many references in that paper.

It is hard to speculate how an online coupling between aDGVM and an ESM would influence simulated vegeta-

130

- tion dynamics due to the non-linearity of feedback mechanisms and the spatially differentiated nature of such effects that will vary between different types of ecosystems. We can therefore only provide rather speculative answers to that question. The work of Zeng et al. (2004) suggests that multiple equilibrium states are possible in semi-arid areas, with grasslands vs. desert being alternative stable states. They also suggest that the range of parameter space over which these equilibria can coexist may be increased by positive feedbacks of evapotranspiration on precipitation (e.g., Wang 135 and Eltahir, 2000). Zhu & Zeng (2014) evaluated the difference between offline and online simulations, but vegetation in their simulations was prescribed and therefore could not respond to climate change. In line with Zhu and Zeng (2014), we would expect that in particular albedo effects, canopy transpiration and evaporation, and temperature effects mitigated by vegetation could alter local to regional climate, in turn feeding back on vegetation dynamics. Where
- 140 and non-linear tipping behavior between different vegetation states could be even more pronounced, because stability is likely enhanced by feedback mechanisms that foster such stability. For example, tropical forests that transfer large quantities of water vapor to the atmosphere via transpiration locally create clouds and precipitation that sustain the existence of such forests even if regional-scale precipitation patterns without such feedbacks showed decreasing trends (see, e.g., Staal et al., 2018). In that sense, such forests foster climatic conditions that sustain their existence. However,

such two-way feedback mechanisms between vegetation and climate exist, we would expect that lag times, bi-stability

- 145 even fully coupled ESMs may be unable to consistently predict how future feedbacks between vegetation and climate will shape terrestrial vegetation state, as shown by Bathiany et al. (2014) in the context of future Sahel greening trends simulated by three different ESMs with dynamic vegetation. We have added a paragraph on this topic to the "limitations" discussion section (p.18 lines 11-34).
- Broadly I like this paper and I think with some minor adjustments, it is suitable for publication. I am very happy to see 150 any revised manuscript version.

**Thank you.**

Small additional things

The Abstract feels a bit too technical in places e.g. use of word "Euclidean". 155

We have rephrased the sentence with the first occurrence of the term "Euclidean distance" to make it more clear that this is used as a measure of dissimilarity between vegetation states. i.e., the sentence "Euclidean distance between simulated transient and equilibrium vegetation states based on selected variables was used to determine lag times and similarity of vegetation states" has be rephrased as follows (p.1 lines 22-27): "We determined lag times and dissimilarity

160 between simulated and transient vegetation states based on the combined difference of 9 selected state variables using Euclidean distance as a measure for that difference." We have further replaced the term "Euclidean distance" with

**"dissimilarity" in lines 39/40 and 47/48.**

165

170

review changes.

Figure 1 (and maybe similar elsewhere). The fonts of the labels and the legends appear very small. One possibility to make more space – at least in the vertical direction – could be to only mark the "x"-axis labels under panels g,h,i.

Thank you for pointing this out. We have altered this figure (p.6) and increased font sizes as much as possible to ensure that labels and legends are more easily legible. The same has been done for all other figures (including supplementary figures) where we now pay attention to have font sizes comparable to the ones used in the main text or the figure captions of the manuscript. To ensure comparability of font sizes and to give an idea of how figures will look like when published, we have used the final journal layout style in the revised manuscript version with the mark-up of the

Figure 3 – the colourbar levels look slightly odd. It feels to me as if they would be neater if simply 0.0, 1.0, 2.0, 3.0, ...

**175 We have re-made the figures in the main text and the supplementary material with colorbars that have even breaks.**

Please check through again in general the diagrams. For instance, I realise it is obvious, but the convention in Figure 4 would be "biomes types are as annotated in panel a. The colours used are common between all four panels".

Thank you for pointing this out, we have changed the figure captions according to your suggestion (p.9, p.12.)

180

Figure 8, with the small font used in the map annotations, it took me some time to realise that the "t" and "e" mentioned in the caption to Figure 8 was added to the end of those annotations. Hence e.g. "RCP8\_5e". Please improve the presentation of this diagram, along with the caption and the annotations.

We have increased the font size of the legend as far as possible and highlight the caption of the legend in bold to make it easier to read. Additionally, we have changed the figure caption to point out which of the panels are representing transient and which ones are representing equilibrium scenarios. The layout of the figure has been altered to maximize panel size in addition to increasing font sizes (p.13).

**190 2 Author responses to comments of anonymous referee #2**

Responses are highlighted in bold font.

The authors present a theoretical study on possible vegetation changes in Africa for two scenarios of global warming and climate change. They use the sophisticated and well documented aDGVM, a dynamic (but not global) vegetation model that has been developed specifically for grass-tree interaction in tropical ecosystems. The authors convincingly demonstrate that in a global warming scenario, the vegetation composition in Africa will likely change and increasingly deviate from its equilibrium composition, i.e., its composition that is attained, if vegetation would instantaneously follow the changing climate. In this sense, the transient future vegetation state in Africa is supposed to move into 'non-analogue states'. In conclusion, this is

a well written, interesting study. The method is clearly outlined. The results are thoroughly and convincingly discussed. The topic is highly relevant. I am happy to recommend its publication in Biogeosciences in its present form subject to a few small, editorial changes.

Thank you for taking the time and making the effort to read and evaluate our manuscript. We are happy that you found it interesting and worthwhile for publication in Biogeosciences.

205

Minor items: Line 234: Fire 'consistently' enlarges. . . ok, but what about statistical significance? I assume the scatter is just too large to talk about statistical significance. This is more a comment, which the authors might consider, not a critical remark. We intentionally wrote "consistently" instead of "significantly" because we did not test for statistical significance

- 210 when aggregating data for Figure 2. The scatter is indeed very large, as indicated by the plotted standard deviations of the spatial means in Fig. 2. This wide scatter is a consequence of the distinct spatial patterning of Euclidean distance emerging over time that can be seen in Fig. 3 and Fig. S2. It is likely that the difference in Euclidean distance between fire and no-fire scenarios is significant for specific regions where fire strongly drives vegetation dynamics, and this then reflects in the consistent difference of the continental-scale mean, which in itself may not be significant. Based on your
- 215 suggestion, we have conducted t-tests and Kolmogorov-Smirnov tests to test the statistical significance of the differences in means between fire and no-fire scenarios. According to these tests, all means were significantly different with p<0.001. We now mention this on p.7 lines 23-25.

Lines 249 to 252: I had to read these sentences at least twice to fully understand their content. Which variables refer to 220 which percentage? Perhaps a slight re-arrangement of the sentence starting with 28% will cure the problem. It slightly enhances understanding, if the authors more specifically refer to Fig. S3a, instead of Fig. S3 (and if the 'Fig. S3a' were put in closed brackets).

Thank you for pointing out your difficulties with these sentences, as well as the formatting issue with the brackets. We have rephrased these sentences to communicate our point more clearly (p.7 lines 54-73).

225

Line 363: What are these unpublished studies by the co-authors (Kumar and Martens)? Grey literature, PhD theses, to be submitted, or just personal communication?

The study of Kumar et al. is meanwhile published as a discussion article (Kumar, D., Pfeiffer, M., Gaillard, C., Langan, L., and Scheiter, S.: Climate change and elevated CO2 favor forest over savanna under different future scenarios

230 in South Asia, Biogeosciences Discuss., https://doi.org/10.5194/bg-2020-169, in review, 2020.), and the study of Martens et al. has been accepted for publication in Global Change Biology. We have updated the references accordingly (p.4 lines78/79, p.14 lines 33-35, p.17 line 103, p.20 lines 88-92 and lines 67-70).

Figures: The figure captions should be self-explaining as much as possible. Therefore, please, explain the acronyms (SDP in Fig. 2, 3, 4 and CDP in Fig. 5, 6, 7 and the figures in the Supplement)

We have updated all figure captions according to your suggestion to make them self-explanatory.

**3 Author responses to comments of anonymous referee #3**

**240 Responses are highlighted in bold font.**

This study compared the simulated transit and equilibrium vegetation states in Africa from 1970 to 2099 under the RCP4.5 and 8.5 scenarios with and without fire. It aims to investigate the time lags of the climate-vegetation system between transit and equilibrium simulations. I think this study is valuable to understand the possible future tipping points of the climate-vegetation

system. I support the publication of this paper after the following comments being addressed properly.

Line 133-135. As for the forcing data, I may support using the original climate forcing even it may produce the saw-tooth pattern. Because the atmospheric internal variabilities could be changed after the randomization. Is the yearly climate forcing given every 10 years or at random frequency? I suppose it is randomized as the former way. If this is true, I think this may not a big problem for the conclusions since the decadal-averaged results are analyzed. But still, I think it's more reasonable to supply the original climate forcing.

250

245

The yearly climate forcing for the spin-up was assembled as a random sequence of the annual climates for the years within a given decade, i.e., the climate of this decade was broken into ten annual blocks, which were then randomly put together to create the 250-year climate sequence for the spin-up. Given that the spin-up was only for one decade and that

- 255 we analysed the output data as decadal-averages, we do not deem possible breaks in climate from one year to the next as problematic. For longer spin-up sequences, e.g., a century of climate data, which may have a significant trend included, the preferred approach would be to de-trend the sequence, then break the de-trended sequence into larger blocks, e.g., decadal blocks or longer, and then randomly assemble these blocks to create the spin-up sequence. This way, the interannual variability is preserved while the trend is removed, and the potential climatology hiatus around the edges of
- 260 the blocks is reduced due to the larger size of the constituting blocks. We have rephrased our explanation on how we

**assembled the spin-up climatology to make it more clear that we used randomized decadal climatology (p.4 lines 49-57).**

Line 171-173. How to define the criteria for the residual distance as close-to-zero and non-zero? Also, I think part of the non-zero residual distance could be caused by the internal variabilities of the climate forcing. It would be better excluding this part properly.

265

I'm not sure I understand what you mean by "excluding this part properly"? In order to test whether the Euclidean distance between transient and equilibrium decade vegetation states is significantly different from zero, one would need to have another reference for comparison in order to determine a typical threshold value. The way to obtain such a reference would be to conduct several equilibrium simulations per decade and scenario, as well as several transient

- 270 simulations per scenario, each with different initializations and, in the case of the equilibrium runs, differently randomized climate year sequences. This would allow to determine the Euclidean distances among the decadal replicates, which then could be compared to the mean Euclidean distance between transient and equilibrium decadal replicates. If the mean Euclidean distance among decadal replicates is statistically significantly smaller than the mean Euclidean distance between transient and equilibrium decadal replicates, then one could quantitatively say the transient-equilibrium
- 275 distance is different from zero. However, due to the large number of simulations already required for this study, we did not conduct replicate simulations that would allow us to directly make such a quantitative statement. Yet, based on experience we know that the between-replicate variability of the state variables used to calculate the Euclidean distance in this study is usually a few percent at best, due to stochasticity between differently initialized runs. Therefore, as a best estimate, we altered our original simulation values, letting them range between  $\pm$  5% difference from the actual
- simulation values in order to mimic typical between-replicate variability. We then, in accordance with the procedure applied to the original variables, standardized the altered variables in the same way. After that, we then, for each grid pixel, each scenario, and each time slice (i.e., decade), calculated the Euclidean distance between original variable tuple and altered variable tuple. This delivered a total of 887848 Euclidean distance values overall that (artifically) represent the typical between-replicate Euclidean distance range. The mean value of this sample was 0.13 ± 0.06, the 95% percentile 0.23, and the 99% percentile was 0.29. It is therefore fairly safe to assume any Euclidean distance > 0.29 is larger
- than zero. We have added this explanation of how we derived an estimate of the non-zero limit to the supplementary material and refer to it in the main text of the manuscript (p.5 lines 27/28 and lines 31-33).

The legend of figure 1 is not clear to distinguish the equilibrium vs. transient on the printed pages. Suggest adding the marks of dot and square on the legend.

We have increased the size of the symbols in Fig. 1 and S1, and changed the legend according to your suggestion (p.6).

The words of "a" in line 201, and "to" in line 598 are written twice. And there is a typo for the word of "both" in line 278.

**4 Author responses to comments of Associate Editor Martin De Kauwe**

**Responses are highlighted in bold font.**

300

I have now received two reviews of your manuscript, both reviewers are positive about your manuscript. I think it is a very interesting study on an important topic and I am recommending minor revisions before publication, many congratulations.

I have read through your responses document and I'm happy with your suggested revisions, so look forward to reading your revised manuscript.

305 A few minor comments:

- In the abstract, when you say "between 1970 and 2099 for the RCP4.5 and 8.5 scenarios", can you please indicate this is based on a single GCM? I would take this sentence to imply aDGVM was forced by the entire CMIP ensemble, for example. I don't mean this comment in any negative sense to be clear, I'm fine with your experimental set up.

**Thank you for your suggestion, we have added a statement specifying that we used regionally downscaled climatology 310 based on the MPI-ESM output for CMIP5 (p.1 lines 20-22).**

- Also, when you talk about lags in the abstract, could you indicate in brackets a couple of examples of what you mean by lags (causes) to capture interested readers? You currently use the word lag(s) seven times, but you don't really explain what is meant. I note you do mention fire at the end, but I still think it would be useful.

315 Good point. We have added a listing of some examples for delays in vegetation response to the second sentence of the abstract (", e.g., changes in physiological processes, structural changes, and changes in vegetation composition and disturbance regimes may happen with substantial delay after a change in forcing has occurred." (p.1, lines 5-8)

When you state: "For example, CO2 fertilization effects may be reduced by increased drought due to water limitation effects
 on plant growth (Temme et al., 2019).". I'm struggling a little with this assertion, firstly because drought is typical short-term for many/most ecosystems, so what evidence is there that it would significantly alter the plants' capacity to exploit a higher CO2 concentration over some multi-year period? Also, if CO2 = greater non-structural carbohydrates or reduces evaporation, both could reduce/delay drought impacts. It is of course also true, that this may make little difference, particularly if the drought is long-lasting, but still, I think more care is needed with this sentence.

325 You are right, the phrasing of this example and the term "drought" is maybe misleading. What we had in mind is changing precipitation regimes in the wake of climate change, for example change in precipitation seasonality (prolonged dry season duration/later start of the wet season), changes in the precipitation frequency distribution, and changes in annual precipitation quantities. Such changes are very likely and actually already observed in different parts of Africa at present (Batisani & Yarnal, 2010; Dunning et al., 2018). While drought may be of short-term du-

- ration (but can last for longer as well, e.g., mega-drought events), changes in precipitation regime that are linked to climate change/changes in atmospheric circulation patterns will have a longer-lasting impact on plant water availability and therefore the capacity of plants to benefit from CO2 fertilization and increased water use efficiency. Where water stress occurs more frequently, e.g., due to increased drought frequency and severity or changes in precipitation seasonality, its negative effects may not automatically offset the beneficial effects of elevated CO2 (see, e.g., Jin et al., 2017: Liu et al. 2020). We have expanded and rephrased this passage to make it more clear what we mean (p.2 lines)
- 335 2017; Liu et al, 2020). We have expanded and rephrased this passage to make it more clear what we mean (p.2 lines 88-104).

Finally, I wonder whether you'd consider adding further to your discussion. You're under no obligation here, just a thought. I think it would be useful to discuss (briefly) key process lags that the model doesn't capture, but in reality may be important.
For example, drought legacy/recovery (I presume it is largely instantaneous in the model - i.e. the model doesn't simulate embolism). You also talk about tree cover decline by the end of the century, I wonder how much acclimation capacity the model has? Can (or does) the model project greater root investment under eCO2? Are greater roots then linked to greater water uptake potential in the model? I guess overall what I'm wondering is how much capacity our models (or your specific model) has to dynamically respond to the projected climate and how much of what we learn is still limited by our model process capacity?

- Yes, the aDGVM ("adaptive DGVM", hence the letter "a" in the model name abbreviation) is able to acclimate to changing environmental conditions. Carbon allocation is dynamic and carbon investment to biomass pools adjusts dynamically in such a way that allocation to those biomass pools that are the most limiting factor for plant growth at a given time is maximized. I.e., if water becomes limiting, plants allocate more carbon to roots at the expense of allocation to other compartments such as stems or leaf biomass. This has been briefly mentioned at the beginning of section 2.1, and we have now added two more sentences to highlight how dynamic allocation works in aDGVM (p.3 lines 66-73). We
- did not specifically keep track of root investment changes over time under eCO2 in this study. This model version does not explicitly simulate xylem cavitation, but can capture drought-related mortality indirectly via carbon-related mortality. If carbon gain is reduced due to water limitation, the carbon balance of a simulated plant individual can become negative (due to respiratory costs exceeding assimilation gains). Negative carbon balance then increases an individual's
- mortality probability. However, what this model version cannot capture yet are more detailed shifts in community trait composition that are caused through environmental filtering, for example shifts towards plant individuals with lower SLA and more negative p50 that can tolerate more water stress. This has however been implemented in the successor model version (aDGVM2, Langan et al., 2017), which we did not use in this study due to its higher computational costs. We have included a brief discussion on the representation of water stress-related mortality in the discussion section 4.1
   (p.14 lines 61-71).

Best wishes,

Martin De Kauwe

**Climate change will cause non-analogue vegetation states in Africa and commit vegetation to long-term change**

Mirjam Pfeiffer1, Dushyant Kumar1, Carola Martens1,2, and Simon Scheiter1

1Senckenberg Biodiversity and Climate Research Centre (BiK-F), Senckenberganlage 25, 60438 Frankfurt am Main, Germany

2Institute of Physical Geography, Goethe University Frankfurt am Main, Altenhoeferallee 1, 60438 Frankfurt am Main, Germany

Correspondence: Mirjam Pfeiffer (mirjam.pfeiffer@senckenberg.de)

**Abstract.**

Vegetation responses to changes in environmental drivers can be subject to temporal lags. This implies that vegetation is committed to future changes once environmental 5 drivers stabilize, e.g., changes in physiological processes,

- structural changes, and changes in vegetation composition and disturbance regimes may happen with substantial delay after a change in forcing has occurred. Understanding the trajectories of such committed changes is important as they
- 10 affect future carbon storage, vegetation structure and community composition and therefore need consideration in conservation management. In this study, we investigate whether transient vegetation states can be represented by a timeshifted trajectory of equilibrium vegetation states, or if they
- 15 are vegetation states without analogue in conceivable equilibrium states. We use a dynamic vegetation model, the aDGVM, to assess deviations between simulated transient and equilibrium vegetation states in Africa between 1970 and 2099 for the RCP4.5 and 8.5 scenarios - Euclidean distance
- 20 between simulated transient and equilibrium using regionally downscaled climatology based on the MPI-ESM output for CMIP5. We determined lag times and dissimilarity between simulated equilibrium and transient vegetation states based on the combined difference of nine selected state vari-
- 25 ables was used to determine lag times and similarity of vegetation statesusing Euclidean distance as a measure for that difference. We found that transient vegetation states over time increasingly deviated from equilibrium states in both RCP scenarios, but that deviation was more pronounced in
- 30 RCP8.5 during the second half of the 21st century. Trajectories of transient vegetation change did not follow a "virtual trajectory" of equilibrium states, but represented nonanalogue composite states resulting from multiple lags with respect to vegetation processes and composition. Lag times between transient and most similar equilibrium vegetation 35 states increased over time and were most pronounced in savanna and woodland areas, where disequilibrium in savanna tree cover frequently acted as main driver for dissimilarities. Fire additionally enhanced lag times and Euclidean distance dissimilarity between transient and equilibrium veg- 40 etation states due to its restraining effect on vegetation succession. Long lag times can be indicative of high rates of change in environmental drivers, of meta-stability and nonanalogue vegetation states, and of augmented risk for future tipping points. For long-term planning, conservation 45 managers should therefore strongly focus on areas where such long lag times and high residual Euclidean distance dissimilarity between most similar transient and equilibrium vegetation states have been simulated. Particularly in such areas, conservation efforts need to consider that observed 50 vegetation may continue to change substantially even after stabilization of external environmental drivers.

*Copyright statement.* © Author(s) 2020. This work is distributed under the Creative Commons Attribution 4.0 License.

[revised manuscript text omitted]